# Orthogonal Hierarchical Decomposition for Structure-Aware Table Understanding with Large Language Models

**Bin Cao** [1 2] **Huixian Lu** [1 2] **Chenwen Ma** [1 2] **Ting Wang** [1 2] **Ruizhe Li** [3 4] **Jing Fan** [1 2]

## Abstract

Complex tables with multi-level headers, merged cells and heterogeneous layouts pose persistent challenges for LLMs in both understanding and reasoning. Existing approaches typically rely on table linearization or normalized grid modeling. However, these representations struggle to explicitly capture hierarchical structures and cross-dimensional dependencies, which can lead to misalignment between structural semantics and textual representations for non-standard tables. To address this issue, we propose an Orthogonal Hierarchical Decomposition (OHD) framework that constructs structure-preserving input representations of complex tables for LLMs. OHD introduces an Orthogonal Tree Induction (OTI) method based on spatial–semantic co-constraints, which decomposes irregular tables into a column tree and a row tree to capture vertical and horizontal hierarchical dependencies, respectively. Building on this representation, we design a dual-pathway association protocol to symmetrically reconstruct semantic lineage of each cell, and incorporate an LLM as a semantic arbitrator to align multi-level semantic information. We evaluate OHD framework on two complex table question answering benchmarks, AITQA and HiTab. Experimental results show that OHD consistently outperforms existing representation paradigms across multiple evaluation metrics.

## 1. Introduction

Tables are ubiquitous in scientific reports, financial statements, and business intelligence, serving as a structured medium to organize multi-dimensional information efficiently (Herzig et al., 2020). With advent of LLMs, there has been a significant shift toward automating table understanding and reasoning tasks (Lu et al., 2025; Li et al., 2026; Sui et al., 2024; Liu et al., 2024b). However, while LLMs exhibit remarkable performance on simple tables, they frequently falter when confronted with **complex heterogeneous tables**.

We define these complex tables as structures that deviate from the canonical $N \times M$ grid, characterized by two primary challenges: (1) **Structural Hierarchy**, where multi-level nested headers require a data cell's semantics to be traced back through a chain of ancestral labels (Wang et al., 2024; Zhao et al., 2022); As illustrated in Figure 1 , complex tables often exhibit pronounced *structural hierarchy*. For example, in the table(c), the interpretation of the entry *"other cities"* is semantically incomplete when considered in isolation. A correct understanding requires jointly resolving its hierarchical dependency with the higher-level header *"Heilongjiang Province"*. Without incorporating this ancestral context, *"other cities"* would be ambiguously interpreted as referring to all cities other than Harbin, rather than the intended meaning: all cities within Heilongjiang Province excluding Harbin. (2) **Spatial-Logical Discontinuity**, where the prevalence of irregularly merged cells and offset headers breaks the canonical grid's $N \times M$ indexing. This misalignment ensures that spatial proximity no longer guarantees logical association, rendering traditional coordinate-based row/column scanning ineffective. As shown in the table(d) of Figure 1 , spatial layout alone may induce misleading hierarchical cues. Based solely on vertical positioning, the entry *"details in 2007"* appears to form a parent–child relationship with the header *"year"*. However, this spatial adjacency does not correspond to a valid logical subsumption. Instead, *"details in 2007"* should be interpreted as a logically parallel attribute rather than a subordinate category of *"year"*.

Existing works to bridge the gap between complex table layouts and LLM reasoning generally fall into three paradigms. *Flat linearization* (Zhang et al., 2024b; Wang et al., 2024) often leads to **structural collapse** where the hierarchical dependencies are lost in the one-dimensional sequence. *Pro-*

[1]Zhejiang University of Technology, China [2]Zhejiang Key Laboratory of Visual Information Intelligent Processing, China 310023 [3]University of Aberdeen, UK [4]University of Birmingham, UK. Correspondence to: Jing Fan <fanjing@zjut.edu.cn>.

*Proceedings of the $43^{rd}$ International Conference on Machine Learning*, Seoul, South Korea. PMLR 306, 2026. Copyright 2026 by the author(s).

**(a)**

| Employee Position | Means of transportation | | | |
|---|---|---|---|---|
| | Airplanes | Trains | Ships | Other means |
| Company leadership | Implemented according to the 'Measures for the Management of Business Expenses of Unit Heads' | | | |
| Other employees | economy class | hard seat | standard class | reimbursable with receipt |

**(b)**

| Feature | | Production cycle | Theoretical capacity | Output rate | Number | Level | Special process |
|---|---|---|---|---|---|---|---|
| Product Components | Lampshades | 30 | 2880 | 98.6% | 200 | 1 | atomization |
| | Decorative frames | 25 | 3456 | 95.3% | 250 | 1 | |
| | Frames | 14 | 1900 | 99.1% | 140 | 2 | |
| | Wall mounts | 36 | 4500 | 94.4% | 500 | 3 | |

**(c)**

| Location | | Accommodation Fee Limit Standard | | | Peak Season Fluctuation Standard | | |
|---|---|---|---|---|---|---|---|
| (City) | | Company leadership | Middle management | Other personnel | Middle management | Company leadership | Other personnel |
| Heilongjiang Province | Harbin | Execute in accordance with relevant regulations | 500 | 500 | 600 | Execute in accordance with relevant regulations | 500 |
| | Other cities | | 400 | 400 | 400 | | 400 |
| Zhejiang Province | Hangzhou | | 600 | 500 | 500 | | 500 |
| | Other cities | | 400 | 400 | 400 | | 400 |
| Shanghai | | | 600 | 600 | 720 | | 600 |
| Anhui Province | Hefei | | 600 | 500 | 500 | | 500 |
| | Other cities | | 400 | 400 | 500 | | 400 |
| Beijing | | | 700 | 600 | 720 | | 600 |

**(d)**

| | Percentage for the first half | | | Percentage for the second half | | |
|---|---|---|---|---|---|---|
| | Together | Man | Women | Together | Man | Women |
| Year | | | | | | |
| 1996 | 48 | 32 | 16 | 56 | 40 | 16 |
| 2007 | 67 | 40 | 27 | 69 | 40 | 29 |
| 2016 | 75 | 43 | 32 | 77 | 57 | 20 |
| Details in 2007 | | | | | | |
| Bachelor's degree or above | 55 | 30 | 25 | 53 | 28 | 25 |
| Age greater than or equal to 20 | 42 | 25 | 17 | 45 | 27 | 18 |
| Considering the first two factors comprehensively | 37 | 24 | 13 | 42 | 25 | 17 |

☐ Column_header     ☐ Row_header     ☐ Data

*Figure 1.* Illustration of table complexity and structural diversity. The examples encompass several challenging non-standard layouts. **(a)**: Tables featuring multi-level nested column headers and merged data cells; **(b)**: Tables characterized by deep hierarchical row header structures; **(c)**: Complex instances with simultaneous multi-layer hierarchies in both rows and columns (dual-axis dependency); **(d)**: Tables with flexible header positioning (column headers located in non-top sections) and highly irregular structural topologies.

*grammatic modeling* (Zhang et al., 2025; 2024a) assumes that tables can be perfectly mapped to a flat header–row format. However, this assumption breaks down for non-canonical layouts, particularly those with flexible or non-fixed header configurations. More recently, *logical topology reconstruction* methods have attempted to recover table logic by graph mapping (Tang et al., 2025; Li et al., 2025). However, these methods are mainly driven by **geometry-based heuristics**. They largely ignore interaction between spatial positioning and linguistic semantics. As a result, they struggle to handle flexible or misaligned headers and fail to establish reliable semantic lineages required for deep reasoning.

To address these limitations, we propose the **Orthogonal Hierarchical Decomposition (OHD)** framework. At the core of OHD is the **Orthogonal Tree Induction (OTI)** algorithm, which leverages **spatial-semantic constraints** to further determine the underlying logical relationships among cells based on three cell roles: *column_header*, *row_header*, and *data*. Unlike existing methods that treat a table as a unified grid, OTI independently induces orthogonal row and column trees. This decoupling allows the framework to isolate structural noise and handle irregular layouts by reconstructing the logical hierarchy of each cell separately. To integrate these dimensions, we design a **dual-pathway association protocol**. This protocol symmetrically restores the semantic lineage of each cell. It employs an LLM as a **semantic arbitrator** to synthesize a high-fidelity, structure-aware

representation. Extensive evaluations on two benchmark datasets for complex table QA, **AITQA** and **HiTab**, demonstrate that OHD significantly outperforms state-of-the-art baselines.

Our contributions are summarized as follows:

- We introduce the **OHD** framework, which shifts the paradigm from global grid modeling to orthogonal hierarchical decomposition, effectively mitigating structural collapse in complex tables.

- We propose **OTI** algorithm, which incorporates spatial-semantic synergy to further determine underlying logical relationships among cells, significantly improving robustness against flexible headers and non-canonical layouts.

- We design a **dual-pathway association protocol** that reconstructs multi-layered semantic lineage of cells, enabling LLMs to perform faithful reasoning on heterogenous structures.

## 2. Related Work

Current research in table understanding and Question Answering (Table QA) has transitioned from simple grid parsing to modeling complex heterogeneous tables characterized by hierarchical headers and spatial-semantic decoupling (Zheng et al., 2023; Fang et al., 2024). Existing methodologies generally follow three paradigms: (1) Flat Serialization,

which linearizes tables into Markdown or HTML/JSON (Chen, 2023; Zhang et al., 2024b); However, this often leads to structural collapse by stripping away orthogonal dependencies and multi-level hierarchical lineages. (2) Programmatic Modeling, which aligns tables with relational schemas (e.g., SQL/DataFrames) (Zhang et al., 2024a; Jiang et al., 2023). These methods suffer from normalization bias, struggling with non-canonical layouts such as irregularly merged headers or embedded sub-titles (Wang et al., 2021). (3) Logical Topology Reconstruction, which uses graphs or trees to recover table skeletons (Li et al., 2025; Tang et al., 2025). Despite their progress, these approaches rely heavily on geometry-based heuristics, failing to capture the synergy between spatial positioning and linguistic semantics. Consequently, they remain vulnerable to structural noise and struggle to resolve the intricate multi-layered semantic binding required for faithful reasoning in complex tables.A detailed taxonomy and analysis of these paradigms are provided in the Appendix A.

## 3. Orthogonal Hierarchical Decomposition

To capture logical dependencies in complex tables, we propose the Orthogonal Hierarchical Decomposition (OHD) framework. The input for OHD is a table with its semantic roles for each cell, i.e., column header, row header or data. OHD factorizes the table into two independent, semantically synchronized hierarchical structures: a column tree ($\mathcal{T}_{\text{col}}$) and a row tree ($\mathcal{T}_{\text{row}}$). Our decomposition is guided by the Semantic-Spatial Synergy principle, which uses semantic roles as a foundation to adjudicate cell relations and steer topological generation. By decoupling these orthogonal dimensions, OHD preserves the structural taxonomy and logical lineage of data cells, providing a high-fidelity representation for LLM-based reasoning. The OHD pipeline, from structural induction to semantic arbitration, is illustrated in Figure 2.

### 3.1. Construction Criteria: Semantic-Spatial Synergy

To ensure the topological integrity and semantic consistency of the orthogonal trees, the construction of $\mathcal{T}_{\text{col}}$ and $\mathcal{T}_{\text{row}}$ is governed by the following synergistic principles:

**Principle of Semantic Agency:** The structural role of a cell within a specific hierarchical tree is strictly dictated by its semantic category (i.e., *column_header*, *row_header*, or *data*). We differentiate two modes of semantic agency depending on the tree under construction: In $\mathcal{T}_{\text{col}}$, only column headers are granted logical branching capabilities, enabling them to serve as internal aggregator nodes. Row headers, along with data cells, are treated as atomic terminal units and can only be attached as leaf nodes to the column-header branch. In $\mathcal{T}_{\text{row}}$, conversely, only row headers possess logical branching capabilities, acting as internal nodes. Column

headers and data cells are reduced to atomic leaf nodes, which are attached to the row-header hierarchy. This preserves topological purity and isolates structural noise from the hierarchical backbone.

**Semantic-Spatial Subsumption:** Building upon the defined semantic roles, the induction of an edge between a parent and a child node requires a rigorous alignment of spatial containment and semantic orientation. Spatially, a parent's bounding box must physically subsume the span of its child. Semantically, this containment must represent a valid attribute inheritance or contextual qualification. Logic edges are induced only upon the convergence of both dimensions, which enables the framework to rectify geometric misalignments commonly found in irregular layouts.

**Structural Unidirectionality & Non-branching:** To establish a deterministic logical chain, we enforce a non-branching constraint on data attributes. Once a node is identified as a data cell, its downward search space is immediately terminated. This hard constraint eliminates *spurious nesting*, e.g., footnotes or auxiliary remarks being misallocated under numerical values, ensuring a unidirectional and unambiguous logical path from root headers to data entries.

This synergistic approach allows OHD to resolve the logical depth of each dimension independently, bypassing the need for global alignment assumptions and enhancing the model's robustness against heterogeneous table structures.

### 3.2. Orthogonal Tree Induction

To capture the bi-directional logical dependencies without relying on global alignment, we propose *Orthogonal Tree Induction (OTI)*, which factorizes the table into a column tree $\mathcal{T}_{\text{col}}$ and a row tree $\mathcal{T}_{\text{row}}$ for independent hierarchical resolution. As $\mathcal{T}_{\text{row}}$ is constructed symmetrically to $\mathcal{T}_{\text{col}}$ by swapping row/column roles (adopting column-major ordering and column-span proximity), we focus on the column tree ($\mathcal{T}_{\text{col}}$) to illustrate this two-step module, detailing its core components: **Header Hierarchy Induction** and **Adaptive Data Anchoring**. The overall procedure of OTI is summarized in Algorithm 1.

**Step 1: Header Hierarchy Induction (line 1-8).** This step aims to uncover the multi-level hierarchical structure among table headers. We first define the column header node set as $\mathcal{H}_{\text{col}}$. For any $h_i, h_j \in \mathcal{H}_{\text{col}}$, start position $s$ and end position $e$, their row spans be $[r_{s,i}, r_{e,i}]$ and $[r_{s,j}, r_{e,j}]$, and their column spans be $[c_{s,i}, c_{e,i}]$ and $[c_{s,j}, c_{e,j}]$, respectively. To order the nodes, we adopt the lexicographic order $<_{lex}$:

$$h_i <_{lex} h_j \iff (r_{s,i} < r_{s,j}) \lor (r_{s,i} = r_{s,j} \land c_{s,i} < c_{s,j}). \tag{1}$$

To establish a valid hierarchical link, a candidate parent node $h_i$ must satisfy the *Spatial-Semantic Judgment ($\mathcal{J}_{ss}$)* with respect to node $h_j$. This relation ensures that the geometric layout aligns with the logical hierarchy through a

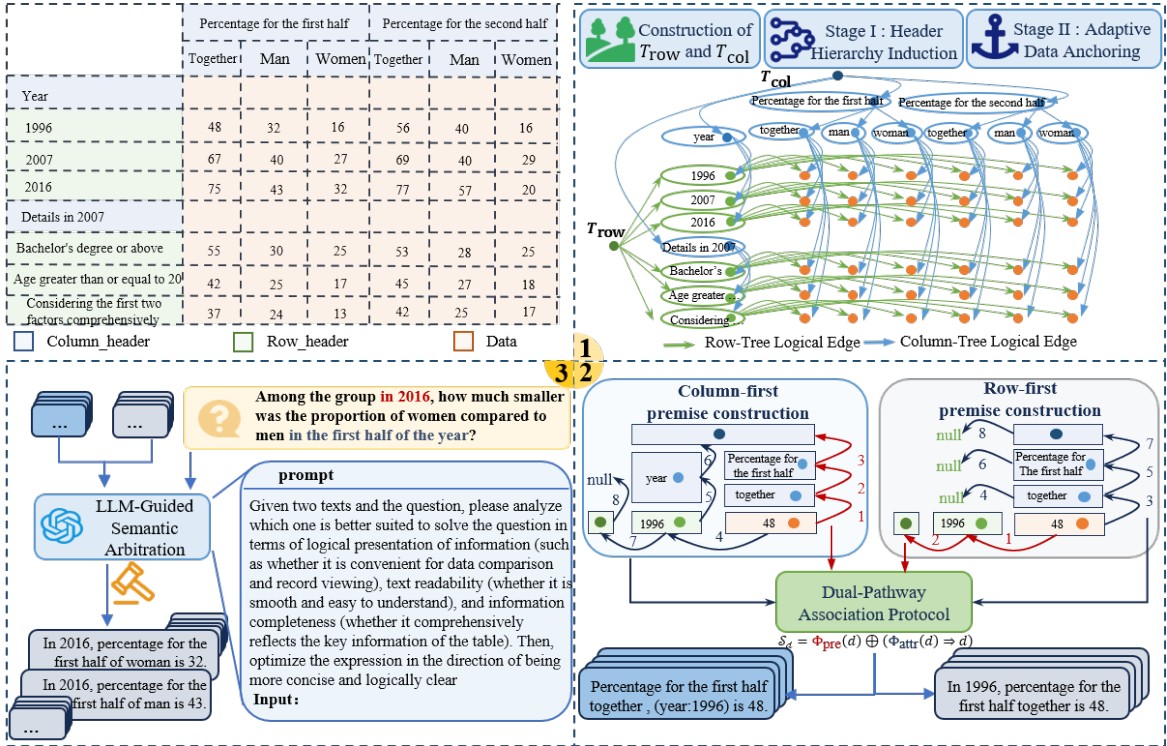

*Figure 2.* Workflow of the OHD framework. The process begins with a Categorized Table Input where each cell is pre-identified as a Row Header, Column Header, or Data unit. The pipeline then proceeds in three stages: (1) Orthogonal Tree Induction (OTI), where we decompose the table into independent row and column hierarchical trees; (2) Dual-Path Association, where we reconstruct the semantic lineage of each cell via synchronized tree traversal; and (3) Semantic Arbitration, where we use an LLM to align hierarchical information into structure-aware prompts.

joint spatial-semantic condition:

$$\mathcal{J}_{ss}(h_j, h_i) \iff \underbrace{[\text{span}_{\text{col}}(h_j) \subseteq \text{span}_{\text{col}}(h_i) \wedge r_{e,i} \leq r_{s,j}]}_{\mathbb{C}_{\text{spatial}}(h_i, h_j)}$$
$$\wedge\, \mathbb{C}_{\text{semantic}}(h_i, h_j)$$
(2)

where $\mathbb{C}_{\text{spatial}}$ and $\mathbb{C}_{\text{semantic}}$ are spatial and semantic constraints respectively. Specifically, $\mathbb{C}_{\text{spatial}}$ requires that the child header $h_j$ is horizontally contained within the column span of the parent header $h_i$, i.e., $\text{span}_{\text{col}}(h_j) \subseteq \text{span}_{\text{col}}(h_i)$ and that $h_i$ is located above $h_j$ in the grid ($r_{e,i} \leq r_{s,j}$), ensuring a natural top-down order. $\mathbb{C}_{\text{semantic}}(h_i, h_j) = 1$ if a large language model determines that a logical subsumption exists between the contents of $h_i$ and $h_j$, e.g., $h_j$ is a subcategory of $h_i$, or $h_j$ is an attribute belonging to $h_i$. This semantic constraint acts as a neural-symbolic bridge that corrects layout ambiguities that spatial heuristics alone cannot resolve. See Appendix C for the specific prompt.

To build the edge set $E$, we define a candidate subset $\mathcal{E}_j = \{(h_i, h_j) \mid \mathcal{J}_{ss}(h_j, h_i), i < j\}$ for each node $h_j$, which consists of all edges satisfying the row-span subsumption relation. Since multiple preceding nodes $h_i$ may belong to this subset, we resolve this ambiguity to ensure a strict hierarchy by only retaining the single edge from $\mathcal{E}_j$ that minimizes the distance between $r_{s,j}$ and $r_{e,i}$. In practice,

this minimal-distance selection is efficiently achieved by an inverse traversal—scanning the already processed nodes in $\mathcal{V}_{built}$ from last to first and connecting $h_j$ to the first valid $h_i$ encountered in $\mathcal{E}_j$. Finally, the complete tree edge set $E$ is obtained by taking the union of these unique parent-child edges selected for all nodes, formally expressed as $E = \bigcup_j \{e_j \mid e_j \in \mathcal{E}_j \text{ is the retained edge}\}$.

**Step 2: Adaptive Data Anchoring (line 9-12).** The purpose of this step is to extract a suitable context for a given data cell. To this end, we consider this data cell as an anchor to connect with higher tree levels, thereby constructing its probable meaning in the table.

**Conflict Set** $\mathcal{S}_{\text{conflict}}$ is defined to identify pairs of headers that share a common parent node in the logical hierarchy but exhibit spatial subsumption. Formally, the conflict set is defined as:

$$\mathcal{S}_{\text{conflict}} = \{\langle h_a, h_b \rangle \mid \exists h_p \in \mathcal{H}_{\text{col}},\ (h_a, h_p), (h_b, h_p) \in E, \\ \wedge\ \mathbb{C}_{\text{spatial}}(h_a, h_b)$$
(3)

After isolating structural layout anomalies via the conflict set $\mathcal{S}_{\text{conflict}}$, the framework anchors each data unit to its appropriate multi-layer logical coordinates. However, naive spatial alignment fails when a data unit lies on the overlapping boundaries of conflicting headers. Therefore, the set of leaf nodes $\mathcal{D}$ is defined as all individual table cells that do not

**Algorithm 1** Orthogonal Tree Induction (OTI)

---

**Require:** Header set $\mathcal{H}$, Data units $\mathcal{D}$, Semantic constraint $\mathbb{C}_{sem}$

**Ensure:** Hierarchical Tree $\mathcal{T} = (V, E)$

1: Sort $\mathcal{H}_{\text{col}}$ in ascending order based on $<_{lex}$
2: **for** each $h_j \in \mathcal{H}_{\text{col}}$ **do**
3:     Find the latest $h_i \in \mathcal{V}_{\text{built}}$ s.t. $\mathcal{J}_{ss}(h_j, h_i)$
4:     **if** $h_i$ exists **then**
5:         $\mathcal{E}_{\text{col}} \leftarrow \mathcal{E}_{\text{col}} \cup \{(h_i, h_j)\}$
6:     **end if**
7:     $\mathcal{V}_{\text{built}} \leftarrow \mathcal{V}_{\text{built}} \cup \{h_j\}$
8: **end for**
9: **for** each $d \in \mathcal{D}$ **do**
10:     Determine parent $\text{Pa}(d)$ via Eq. 4
11:     $V \leftarrow V \cup \{d\}$, $E \leftarrow E \cup \{(\text{Pa}(d), d)\}$
12: **end for**
13: **return** $\mathcal{T} = (V, E)$

---

belong to $\mathcal{H}_P$. For each leaf node $d \in \mathcal{D}$, we first retrieve all header cells $h_a$ such that the column span of $d$ is fully contained within that of $h_a$, i.e., $\text{span}_{\text{col}}(d) \subseteq \text{span}_{\text{col}}(h_a)$ and the starting row of $d$ lies below $h_a$ ($r_{e,a} > r_{s,d}$). Then, to resolve cross-layer ambiguity, we further examine whether a conflicting header pair $\langle h_a, h_b \rangle \in \mathcal{S}_{\text{conflict}}$ exists to determine the true parent of $d$. In this specific scenario, the exact parent header $\text{Pa}(d)$ for each data unit $d \in \mathcal{D}$ is adaptively determined by computing row-boundary constraints.

$$\text{Pa}(d) = \begin{cases} h_a, & \text{if } \exists \langle h_a, h_b \rangle \in \mathcal{S}_{\text{conflict}} \wedge r_{e,d} < r_{s,b} \\ h_b, & \text{if } \exists \langle h_a, h_b \rangle \in \mathcal{S}_{\text{conflict}} \wedge r_{s,d} > r_{e,b} \\ h_a, & \text{otherwise} \end{cases} \quad (4)$$

Equation 4 ensures robust anchoring in heterogeneous layouts. A concrete scenario is exemplified by the *Year* ($h_a$) and *Details in 2007* ($h_b$) column headers in Figure 2, where both branch from the same parent but overspread unevenly in the layout. Specifically, when anchoring the data cell *2016* ($d$), the framework triggers the conflict pair $\langle h_a, h_b \rangle \in \mathcal{S}_{\text{conflict}}$. Since its vertical coordinate satisfies the upper-bound constraint ($r_{e,d} < r_{s,b}$), Equation 4 bypasses the sub-header $h_b$ and adaptively assigns the macro-header *Year* ($h_a$) as its correct parent node $\text{Pa}(d)$.

### 3.3. Dual-Pathway Association

After obtaining the $\mathcal{T}_{\text{row}}$ and $\mathcal{T}_{\text{col}}$ from OTI, this stage focuses on reassembling the scattered headers and data cells into a coherent piece of text. We design a **dual-pathway association protocol**: we choose one direction as the premise context, and the other orthogonal direction as the attribute context. For each data cell $d$, we generate a structured text according to the following formula:

$$\mathcal{S}_d = \Phi_{\text{pre}}(d) \oplus (\Phi_{attr}(d) \Rightarrow d) \quad (5)$$

here $\Phi_{\text{pre}}(d)$ is the path along the first direction, serving as the premise context; $\Phi_{\text{attr}}(d)$ is the path along the orthogonal direction, serving as the attribute context. This results in a clear "Premise Context → Attribute Context → Value" structure.

To fully preserve the structural details of the table from two complementary perspectives, OHD performs construction using $\mathcal{T}_{\text{row}}$ and $\mathcal{T}_{\text{col}}$ separately as the first tree. Let $\mathcal{T}_P \in \{\mathcal{T}_{\text{col}}, \mathcal{T}_{\text{row}}\}$ be the first tree, i.e., the first direction, and $\mathcal{T}_O$ be the corresponding orthogonal tree, i.e., the other direction. For each perspective, the dual-pathway protocol described above generates a representation. The overall procedure of dual-pathway association is summarized in Algorithm 2.

**Step 1: Premise Context Construction (line 4-5).** We perform a depth-first search (DFS) on the first tree $\mathcal{T}_P$, which produces an ordered sequence of header nodes:

$$\mathcal{Q}_P = [\, h \mid h \in \text{DFS}(\mathcal{T}_P),\ h \in \mathcal{H}_P \,]. \quad (6)$$

During the traversal, whenever we encounter a header node, we add it to the current path as premise context for subsequent data nodes. When we reach a data node $d$, the sequence of header nodes along the path from the root to $d$ becomes its premise context $\Phi_{\text{pre}}(d)$. In other words, the DFS simultaneously determines the order in which data cells are visited as given by $\mathcal{Q}_P$ and constructs the premise context for each data cell.

**Step 2: Attribute Context Construction (line 6-7).** After establishing the premise context, we extract the context of the structural attribute by traversing $\mathcal{T}_O$ in a bottom-up manner. For each data node $d$, we trace the unique ancestral path from $d$ up to the root of $\mathcal{T}_O$, denoted as $\text{Path}^{\uparrow}_{\mathcal{T}_O}(d) = \langle h_1, h_2, \ldots, h_k \rangle$, where $h_1$ is the immediate parent header of $d$ and $h_k$ is the root header in the $\mathcal{T}_O$ dimension.

To resolve cross-dimensional layout ambiguities, as the model encounters each header $h_i \in \text{Path}^{\uparrow}_{\mathcal{T}_O}(d)$ during this upward traversal, it queries $\mathcal{T}_P$ via an inverse retrieval to capture all corresponding headers in the opposite dimension, denoted as $\text{Anc}_{\mathcal{T}_P}(h_i)$. This complementary contextual lineage for each header is aggregated into a sub-sequence $\text{Seq}(\text{Anc}_{\mathcal{T}_P}(h_i))$. Formally, the complete attribute context $\Phi_{\text{attr}}(d)$ is finally presented and constructed as follows:

$$\Phi_{\text{attr}}(d) = \bigoplus_{h_i \in \text{Path}^{\uparrow}_{\mathcal{T}_O}(d)} \left( \text{Seq}(\text{Anc}_{\mathcal{T}_P}(h_i)) \oplus \langle h_i \rangle \right) \quad (7)$$

where $\oplus$ denotes the sequence concatenation operator. By interleaving the ancestral paths of both orthogonal hierarchies during the upward climbing of $\mathcal{T}_O$, $\mathcal{S}_d$ fully recovers the multi-dimensional semantic coordinates of each individual data cell.

**Step 3: Sequential Concatenation (line 8).** Let $n$ denote the current node during a depth-first search (DFS) of the first

**Algorithm 2** Dual-Pathway Association

---

**Require:** Trees $\mathcal{T}_{\text{col}}, \mathcal{T}_{\text{row}}$, Data cells $\mathcal{D}$
**Ensure:** Integrated sequences $\mathcal{R}_{\text{col}}, \mathcal{R}_{\text{row}}$
 1: **for** each $\mathcal{T}_P \in \{\mathcal{T}_{\text{col}}, \mathcal{T}_{\text{row}}\}$ **do**
 2:     $\mathcal{R}_P \leftarrow \langle\rangle$
 3:     **for** each $n \in \text{DFS}(\mathcal{T}_P)$ **do**
 4:         **if** $n \in \mathcal{H}_P$ **then**
 5:             $\mathcal{R}_P \leftarrow \mathcal{R}_P \oplus \langle n \rangle$ {Premise Context}
 6:         **else** $\{n \in \mathcal{D}\}$
 7:             Get $\Phi_{\text{pre}}(n)$ from $\mathcal{T}_P$ path, and $\Phi_{\text{attr}}(n)$ via Eq. (7) on $\mathcal{T}_O$
 8:             $\mathcal{R}_P \leftarrow \mathcal{R}_P \oplus \Phi_{\text{pre}}(n) \oplus (\Phi_{\text{attr}}(n) \Rightarrow n)$
 9:         **end if**
10:     **end for**
11: **end for**
12: **return** $\mathcal{R}_{\text{col}}, \mathcal{R}_{\text{row}}$

---

tree $\mathcal{T}_P$. The final context $\mathcal{R}_P$ is synthesized by combining the premise context, attribute context, and the data cell:

$$\mathcal{R}_P = \bigoplus_{n \in DFS(\mathcal{T}_P)} \begin{cases} n, & n \in \mathcal{H}_P, \\ \Phi_{\text{pre}}(n) \oplus (\Phi_{\text{attr}}(n) \Rightarrow n), & n \in \mathcal{D}. \end{cases} \tag{8}$$

### 3.4. Semantic Arbitration

The final stage of our framework involves **Multi-pathway Semantic Arbitration**. Recognizing that $\mathcal{T}_{\text{col}}$ and $\mathcal{T}_{\text{row}}$ offer complementary topological views, we leverage Large Language Models (LLMs) to flexibly and adaptively select the clearest perspective tailored to the specific question. This ensures the model always reasons through the optimal structural axis, avoiding the limitations of a single, fixed perspective.

**Arbitration Criteria.** The LLM serves as a semantic arbitrator, evaluating the synthesized candidate sequences across three pivotal dimensions: (1) *Logical Cohesion:* Ensuring the hierarchical nesting of attributes is accurately reflected without semantic fragmentation. (2) *Information Completeness:* Verifying that key data anchors from both orthogonal views are comprehensively integrated. (3) *Syntactic Readability:* Refining the flow of natural language to transform structural fragments into coherent narratives.

To guide the LLM, we design a zero-shot prompt $\mathcal{I}$. The final refined sequence $\mathcal{S}_{\text{final}}$ is obtained by: $\mathcal{S}_{\text{final}} = \text{LLM}(\mathcal{R}_{\text{col}}, \mathcal{R}_{\text{row}}, \mathcal{I})$. The prompt explicitly asks the model to be concise and logically clear. By blending the strengths of column-first and row-first contexts, this process produces a high-fidelity textual surrogate of the original semi-structured table, ready for complex downstream reasoning tasks. See Appendix D for the specific prompt.

## 4. Discussion and Case Study

While the previous sections establish the theoretical and algorithmic foundation of **OHD**, this section provides a more granular examination of its efficacy through a series of complex, real-world structural challenges. By subjecting our framework to specific queries derived from the heterogeneous layouts in Figure 1 across two special questions, we intuitively demonstrate how OHD navigates the pitfalls of structural ambiguity that frequently mislead conventional parsers.

*Question 1: In how many provinces or municipalites does the peak-season standard for vocational secondary school employees exceed 450 yuan in the table(c) of Figure 1 ?*

Thanks to the proposed **OTI**, our method correctly recovers hierarchical administrative relations, such as those between Heilongjiang Province, Harbin, and other subordinate cities. Consequently, when determining whether the peak-season standard exceeds 450 yuan, our approach performs value comparisons across all relevant entries, e.g., *600, 400* versus *450*, leading to the correct conclusion that Heilongjiang Province does not satisfy the condition.As a result, our method identifies exactly three valid regions—*Shanghai, Anhui Province, and Beijing*. In contrast, flattening-based baselines fail to preserve inter-city dependencies within provinces, e.g., *other cities* in *Heilongjiang* and *Zhejiang*, which causes incomplete comparisons and false positives, ultimately producing an incorrect count of five.

*Question 2: What is the percentage of the population together with education below a bachelor's degree in the first half of 2007 in the table(d) of Figure 1?*

We analyze this query regarding population data from the *"percentage of the first half "* in Table (d) of Figure 1 . The primary challenge is a layout artifact where *"details in 2007"* is physically nested underneath the *"2016"* header. In our framework, because the row and column hierarchies are constructed independently, *"2016"* (identified as a data) is strictly prohibited from possessing branching capabilities during the induction of the column tree $\mathcal{T}_{\text{col}}$. This architectural constraint inherently prevents any erroneous association between the two entries, effectively decoupling the deceptive spatial proximity. Consequently, the model avoids misattributing *2007* data to the *2016* hierarchy, enabling the precise extraction of the target value (**55**). By executing the subsequent calculation ($67 - 55$), OHD yields the correct final result of 12. In contrast, conventional methods suffer from **structural occlusion**, as they fail to distinguish between orthogonal header roles within a unified grid. The 2007 details remain "hidden" within the 2016 layer due to rigid coordinate-based parsing, forcing baselines to rely on the previously indexed aggregate value (**67**) and leading to an erroneous conclusion.

# 5. Experiments

## 5.1. Datasets

We evaluate our framework on two public complex table question answering benchmarks: **AITQA** (Katsis et al., 2022) and **HiTab** (Cheng et al., 2022). Both datasets contain multi-level headers, merged cells, and irregular layouts that pose significant challenges for table serialization and reasoning. AITQA consists of financial and statistical tables with nested headers and cross-row dependencies. HiTab focuses on hierarchical tables requiring multi-hop reasoning over row and column structures. Considering the architectural focus of our framework on tables of moderate scale, we curate a refined HiTab subset by imposing a dimensionality constraint that limits either dimension to at most 50.

## 5.2. Baselines

To ensure a comprehensive evaluation, we categorize our baselines according to the three structural paradigms established in our related work: **1) Flat Serialization-based Baselines:** This group represents the mainstream approach of treating table understanding as a sequence-modeling task. E5 (Zhang et al., 2024b) is an embedding-optimized serialization framework designed to enhance semantic retrieval within tables. Chain-of-Table (Wang et al., 2024) is an advanced reasoning-centric baseline that performs stepwise decomposition over Markdown tables. **2) Schema-Alignment Baselines:** We select TableLlama (Zhang et al., 2024a), which represents the programmatic paradigm. It utilizes instruction tuning to align tables with canonical relational schemas. **3) Logical Topology Reconstruction Baselines:** We benchmark against ST-RAPTOR (Tang et al., 2025), a pioneering graph-based method. Unlike our semantic-driven approach, it reconstructs logical edges primarily via geometric layout features, e.g., visual borders.

## 5.3. Evaluation Metrics

Following established benchmarks (Zhao et al., 2023; Zheng et al., 2023; Zhang et al., 2024b), we employ a multifaceted evaluation protocol. First, Exact Match (EM) is reported to assess the model's precision in generating strictly correct outputs, without any tolerance for lexical variation. Second, to account for semantic variability beyond surface-level string matching, we utilize an LLM-based evaluator (LLM Eval). To ensure robustness and mitigate model-specific bias, our LLM Eval aggregates the averaged judgments from three diverse large language model backends: Qwen2-72B (Team et al., 2024), DeepSeek-v3 (Liu et al., 2024a), and GPT-4 (Baktash & Dawodi, 2023). This ensemble approach, by leveraging multiple independent evaluators, yields a stable and fair assessment of semantic accuracy, reducing the risk of overfitting to any single judge's preference.

## 5.4. Main Results

Table 1 summarizes the performance of our OHD framework against representative baselines across multiple datasets. Our method demonstrates consistent and substantial gains across different LLM backbones.

Utilizing Qwen2-72B, OHD achieves an EM score of 69.34 on AITQA, outperforming the strongest baseline (St-Raptor) by 8.76 absolute points. The improvement is even more significant in LLM Eval Avg., where OHD reaches 89.12 (+18.09 over St-Raptor). On the complex HiTab and its Subset, OHD maintains a clear lead with EM scores of 60.16 (+6.31) and 64.74 (+9.01), respectively. These results suggest that orthogonal hierarchical representations effectively resolve semantic ambiguities in complex tables, fostering more precise reasoning. Evaluated with TableLLaMA-7B, OHD achieves a peak EM of 73.78 on AITQA and an incremental gain of 1.62 points on the HiTab Subset. On the full HiTab set, OHD's EM (63.62) slightly trails the vanilla backbone (64.71) due to context truncation caused by the token overhead of explicit structural decomposition. These results suggest that while explicit processing supplements implicit fine-tuning, its effectiveness requires balancing representation completeness against sequence length constraints.

## 5.5. Ablation Study

To investigate the individual contribution of each component within the OHD framework, we conduct extensive ablation experiments on the AITQA and HiTab datasets.

The variants are categorized into three dimensions: **1) Structural Induction Constraints:** The variant *w/o* $\mathbb{C}_{semantic}$ disables the semantic correction mechanism in the Orthogonal Tree Induction (OTI) process in Section 3.2, relying solely on geometric spatial relationships for tree construction. **2) Dual-Pathway Association Strategy:** The variants *w/o* $\mathcal{T}_P$ ($\mathcal{T}_{col}$) and *w/o* $\mathcal{T}_P$ ($\mathcal{T}_{row}$) restrict the Structural Association Reconstruction in Section 3.3 to a single primary axis. The variants *w/o Association (Markdown/HTML)* and *w/o Association (HTML)* replace the proposed dual-pathway hierarchical path representation with standard table serialization formats, specifically markdown tables or HTML code, to evaluate the effectiveness of our logical structural grounding against traditional flat sequences. **3) Semantic Arbitration:** The variant *w/o LLM-based Heuristics* bypasses the arbitration stage in Section 3.4 by directly concatenating both serialized pathways as input.

The ablation results, summarized in Table 2, quantify the individual contributions of OHD's core components to its overall reasoning performance. By systematically deconstructing the framework, we observe several critical insights into how orthogonal hierarchical decomposition facilitates complex table understanding:

*Table 1.* Main performance comparison on AITQA and HiTab benchmarks. This table reports the results of the OHD framework using Qwen2-72B and TableLLaMA-7B as backbones, compared against competitive baselines including Chain-of-Table, E5, and ST-RAPTOR. Evaluation metrics comprise Exact Match (EM) and an LLM-based holistic score (LLM Eval Avg.) to reflect reasoning quality. Bold values indicate the best performance under the same backbone configuration.

| Method | AITQA | | HiTab | | HiTab Subset | |
|---|---|---|---|---|---|---|
| | EM | LLM Eval Avg. | EM | LLM Eval Avg. | EM | LLM Eval Avg. |
| Chain-of-Table (Qwen2-72B) | 49.32 | 62.02 | 44.26 | 62.92 | 46.09 | 64.06 |
| E5 (Qwen2-72B) | 56.31 | 58.97 | 43.56 | 47.93 | 44.49 | 48.78 |
| St-Raptor (Qwen2-72B) | 60.58 | 71.03 | 53.85 | 60.71 | 55.73 | 61.91 |
| **Ours (Qwen2-72B)** | **69.34** | **89.12** | **60.16** | **67.15** | **64.74** | **70.66** |
| TableLLaMA-7B | 68.35 | 85.61 | **64.71** | **66.99** | 66.75 | 71.56 |
| **Ours (TableLLaMA-7B)** | **73.78** | **87.95** | 63.62 | 66.24 | **68.37** | **74.23** |

*Table 2.* Detailed ablation study of the OHD framework on AITQA and HiTab datasets using the Qwen2-72B backbone. The components evaluated include: (1) **Semantic Constraint** ($\mathbb{C}_{\text{semantic}}$), representing spatial-semantic co-constraints; (2) **Orthogonal Pathways** ($\mathcal{T}_{col}, \mathcal{T}_{row}$), demonstrating the necessity of independent axial tree induction; (3) **Dual-Path Association**, comparing our structure-aware representation against conventional Markdown and HTML linearization; and (4) **LLM-based Heuristics**, showing the role of semantic arbitration. Values in parentheses denote the performance degradation compared to the **Full OHD** framework.

| Variant | AITQA | | HiTab | |
|---|---|---|---|---|
| | EM | LLM Eval Avg. | EM | LLM Eval Avg. |
| w/o $\mathbb{C}_{\text{semantic}}$ | 63.30 (−6.04) | 84.24 (−4.88) | 53.72 (−6.44) | 59.90 (−7.25) |
| w/o $\mathcal{T}_P$ ($\mathcal{T}_{col}$) | 49.13 (−20.21) | 68.61 (−20.51) | 56.88 (−3.28) | 62.56 (−4.59) |
| w/o $\mathcal{T}_P$ ($\mathcal{T}_{row}$) | 60.78 (−8.56) | 86.00 (−3.12) | 56.44 (−3.72) | 63.30 (−3.85) |
| w/o Association (Markdown) | 53.20 (−16.14) | 60.08 (−29.04) | 53.35 (−6.81) | 58.20 (−8.95) |
| w/o Association (HTML) | 50.10 (−19.24) | 61.23 (−27.89) | 55.18 (−4.98) | 59.33 (−7.82) |
| w/o LLM-based Heuristics | 68.74 (−0.60) | 86.21 (−2.91) | 59.79 (−0.37) | 65.33 (−1.82) |
| **Full OHD** | **69.34** | **89.12** | **60.16** | **67.15** |

**Effectiveness of Semantic-Spatial Synergy:** The integration of spatial and semantic constraints is essential for robust tree induction. Removing the semantic constraint (**w/o $\mathbb{C}_{\text{semantic}}$**) leads to a consistent decline of 4.88% to 7.25% in EM and LLM-based scores, underscoring the necessity of LLM-driven semantic predicates in resolving structural ambiguities. Specifically, the degradation in this variant on AITQA suggests that geometric proximity alone is insufficient for non-standard layouts, where semantic validation serves as a necessary corrective measure.

**Validation of Structural Integrity through Dual-Pathway Protocol:** The dual-pathway reconstruction demonstrates clear advantages over single-axis or traditional serialization methods. As shown in the results, the absence of the column tree (**w/o $\mathcal{T}_P$ ($\mathcal{T}_{col}$)**) triggers the most substantial performance drop on AITQA (from 69.34% to 49.13% EM), identifying vertical hierarchical dependencies as a vital bottleneck for AITQA, whereas both pathways are highly complementary in capturing the full multi-dimensional semantic association on HiTab. Furthermore, replacing OHD's logical association with standard formats (**w/o Association (Markdown/HTML)**) results in a performance loss of over 10% in EM across most benchmarks. This substantial decline confirms that OHD's hierarchical path representation

provides much richer structural grounding than traditional flat sequences.

**Importance of LLM-based Heuristics in Structural Arbitration:** The **w/o LLM-based Heuristics** variant shows a moderate performance decrease ranging from 0.37% to 2.91% by directly concatenating both serialized pathways as input without selective filtering. This indicates that excessive structural noise from dual-pathways can easily overwhelm LLM's downstream reasoning capacity. It highlights the critical importance of our heuristic-based arbitration in distilling task-relevant context, thereby mitigating structural redundancy and streamlining the reasoning process. Overall, the full OHD configuration achieves the highest scores across all metrics, suggesting that the synergy between orthogonal topology induction and dual-pathway association is critical for handling heterogeneous table structures.

### 5.6. Backbone Generalization and Robustness Analysis

To validate the model-agnostic robustness of the OHD framework and mitigate potential biases in instruction-following capabilities that may arise from specialized table-tuning, we extend our evaluation to include three prominent general-purpose Large Language Models (LLMs) as back-

bones: **Qwen2-72B-Instruct**, **Qwen2.5-72B-Instruct**, and **DeepSeek-V3**.

We evaluate OHD against the **Raw Table QA** baseline, which serves as a direct-serialization paradigm. In this setup, original tables are fed into the LLM without any structural decomposition, utilizing a Markdown-based format. This experimental design allows us to effectively decouple the performance contributions of the underlying LLM from the structural gains provided by the OHD framework. Consequently, it facilitates a rigorous comparison between raw serialized contexts and our structural induction method, demonstrating OHD's superior adaptability and effectiveness across diverse state-of-the-art base models.

*Table 3.* Performance comparison between Raw Table QA (Baseline) and OHD on Qwen2-72B-Instruct, Qwen2.5-72B-Instruct, and DeepSeek-V3.

| Backbone | Method | AITQA | | HiTab | |
|---|---|---|---|---|---|
| | | EM | LLM Eval Avg. | EM | LLM Eval Avg. |
| Qwen2-72B | Raw Table QA | 45.22 | 49.32 | 16.48 | 34.79 |
| | **OHD (Ours)** | **69.34** | **89.12** | **60.16** | **67.15** |
| Qwen2.5-72B | Raw Table QA | 30.29 | 31.46 | 14.14 | 28.35 |
| | **OHD (Ours)** | **72.62** | **84.85** | **66.92** | **88.38** |
| DeepSeek-V3 | Raw Table QA | 38.25 | 42.91 | 15.85 | 29.73 |
| | **OHD (Ours)** | **73.01** | **86.21** | **67.74** | **86.21** |

As summarized in Table 3, OHD achieves substantial improvements over the Raw Table QA baseline across all backbones and metrics. Notably, on the **DeepSeek-V3** model, OHD elevates the EM score on HiTab from a modest **15.85%** to a remarkable **67.74%**. This stark contrast demonstrates that even frontier LLMs encounter significant bottlenecks when processing raw, serialized tabular data. OHD's structural induction acts as a critical bridge for complex layouts, providing the necessary logical scaffolding that raw input lacks. The consistent gains—ranging from **+24 to +52 percentage points** in absolute terms across all datasets and metrics—confirm that OHD is a robust, plug-and-play architecture that complements and amplifies the evolving reasoning capabilities of modern LLMs.

### 5.7. Impact of Table Scale

To address the concern regarding whether OHD maintains its efficacy as table dimensions increase, we conduct a comparative analysis across three scaling tiers defined by the range of row and column counts: **Small** (both rows and columns $< 30$), **Medium** ($30 \leq$ rows or cols $\leq 40$), and **Large** (rows or columns $> 40$). This constraint naturally aligns with our orthogonal tree induction design. By decoupling the row and column hierarchies during construction, applying independent restrictions to each dimension guarantees that the structural decomposition remains computationally tractable and highly focused.

*Table 4.* Performance (Accuracy/EM) comparison across different table scales based on the Qwen2-72B backbone. Table scales are categorized as **Small**, **Medium**, and **Large**. The results demonstrate OHD's superior resilience to structural scaling.

| Method | Small | Medium | Large |
|---|---|---|---|
| Raw Table QA | 17.92 | 13.89 | 10.97 |
| Chain-of-Table | 47.91 | 40.28 | 28.86 |
| E5 | 44.39 | 43.05 | 39.84 |
| ST-Raptor | 57.37 | 47.22 | 40.65 |
| **OHD (Ours)** | **61.56** | **58.33** | **54.47** |

As illustrated in Table 4, while the absolute accuracy of all methods naturally decreases as tables expand, the performance gap between OHD and the weak baseline (Raw Table QA) remains consistently large, from **43.64** on Small to **43.50** on Large. Moreover, OHD outperforms all other strong baselines (e.g., ST-Raptor and Chain-of-Table) by a wide margin across all scales. Notably, while competitive baselines such as ST-Raptor and Chain-of-Table experience sharp performance drops ($-16.72$ and $-19.05$ points, respectively) when shifting from small to large scales, OHD only shows a marginal decline of **7.09** points. This demonstrates that OHD's structural induction provides a more stable and context-efficient representation, effectively mitigating the information loss and reasoning confusion that typically plague standard serialization or geometric-only methods as table complexity scales.

## 6. Conclusion

In this paper, we presented the **Orthogonal Hierarchical Decomposition (OHD)** framework, a novel paradigm designed to bridge the gap between complex two-dimensional table topologies and the linear reasoning capabilities of large language models. By decoupling irregular table grids into independent row and column hierarchical trees through our *Orthogonal Tree Induction* (OTI) algorithm, we successfully transformed fragile physical layouts into robust, structure-aware semantic lineages. Our extensive empirical evaluations on the AITQA and HiTab benchmarks demonstrate that OHD significantly outperforms state-of-the-art linearization and retrieval-augmented baselines, particularly in scenarios involving multi-level nested headers and merged cells. The ablation studies further underscore that the dual-path lineage representation is the key driver of performance, effectively mitigating the structural collapse common in traditional representations. For future work, we aim to extend the OHD framework to handle ultra-large-scale financial reports and explore the potential of integrating multimodal signals (e.g., visual layout cues) to further enhance the robustness of semantic agency identification. We believe that the principle of orthogonal decomposition provides a promising direction for achieving more granular and reliable table understanding in diverse real-world applications.

## Acknowledgement

This research was partially supported by: Baima Lake Laboratory Joint Fund of the Zhejiang Provincial Natural Science Foundation of China (No.LBMHZ25F020001) and the National Natural Science Foundation of China (Grant No. 62276233).

## Impact Statement

This paper presents the Orthogonal Hierarchical Decomposition (OHD) framework, which aims to enhance the structural understanding and reasoning capabilities of large language models for complex tables. The broader social impact of our work is twofold. On the positive side, it facilitates the automation of high-fidelity data extraction and analysis in critical domains such as financial reporting, medical record management, and scientific research, thereby reducing human error and improving decision-making efficiency. On the ethical side, as with any automated reasoning system, there is a potential risk that structural misinterpretations could lead to incorrect conclusions if used in high-stakes environments without human oversight. We encourage practitioners to utilize OHD as a supportive tool rather than a final decision-maker. We believe there are no specific ethical concerns or negative social consequences that require additional highlight beyond these standard considerations.

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

# A. Extended Related Work

The evolution of Table Question Answering (Table QA) has shifted from simple grid-based parsing toward the structural modeling of **complex heterogeneous tables** (Zheng et al., 2023; Fang et al., 2024). Such tables, as exemplified by multi-level hierarchies and non-linear data dependencies, pose a fundamental challenge: preserving structural integrity during model input. We categorize existing methodologies into three primary paradigms and analyze their specific bottlenecks.

**Flat Serialization-based Representations**    This paradigm maps two-dimensional tabular structures into one-dimensional text streams through predefined linearization rules, such as Markdown (Chen, 2023; Liu et al., 2024b; Zhao & Sun, 2024), JSON, and HTML (Zhang et al., 2024b). While these methods leverage the sequence-modeling strengths of Large Language Models (LLMs), they inherently suffer from **structural collapse**. By flattening hierarchical headers and merged cells into a linear string, they strip away the **orthogonal dependencies** between rows and columns. Consequently, the model's ability to trace the semantic lineage of a data cell back to its multi-level ancestors is severely compromised, particularly when the logical depth exceeds the model's contextual window.

**Programmatic Modeling via Schema Alignment**    To introduce relational rigor, this paradigm (Zhang et al., 2024a; Jiang et al., 2023) represents tables as structured objects, such as SQL tables or DataFrames, conforming to canonical relational schemas. However, these methods rely on a **normalization bias**, assuming that tables can be perfectly mapped to a flat relational header-row format. In complex tables in the real-world, unconventional layouts—such as **irregularly merged headers, embedded sub-titles, or empty top-left corner cells**—defy standard normalization (Wang et al., 2021). Force-fitting such heterogeneous structures into rigid schemas leads to a secondary loss of semantic information, making programmatic reasoning fragile when the physical layout is non-canonical (Zhang et al., 2025; Buss et al., 2025).

**Logical Topology Reconstruction**    Recent advances (Li et al., 2025; He et al., 2024; Tang et al., 2025) attempt to recover the skeleton of the table by modeling logical dependencies such as heterogeneous graphs or structural trees. For example, ST-RAPTOR (Tang et al., 2025) identifies geometric physical features to map cell relationships. Despite their progress, current reconstruction processes are predominantly driven by **geometry-based heuristics**, which overlook the **synergy between spatial positioning and linguistic semantics**. These methods struggle with **flexible and misaligned headers** because they lack the capacity to dynamically adjudicate a cell's role based on its content. Furthermore, by treating the table as a unified grid rather than independently inducing orthogonal row and column hierarchies, they remain vulnerable to structural noise and fail to resolve the multi-layered semantic binding required for faithful reasoning.

# B. Benchmarking the Consistency of LLM-based Evaluators

In addition to Exact Match (EM), we adopt an LLM-based evaluation protocol to assess semantic correctness beyond surface-level string matching. To examine the stability and reliability of such evaluations, we employ three different large language models as independent evaluators, namely **Qwen2-72b**, **DeepSeek-v3**, and **GPT-4**. All evaluators are provided with the same evaluation prompt and are required to judge whether a model prediction is semantically consistent with the ground-truth answer. To mitigate the impact of inherent model stochasticity and ensure the stability of our findings, the reported results for each evaluator are averaged over multiple independent trials.

Importantly, this cross-evaluator analysis is conducted consistently across both *baseline comparisons* and *ablation studies*. Specifically, Tables 5–7 report the detailed LLM-based evaluation results for baseline methods on three benchmark datasets, while Table 8 presents the corresponding results for the ablation variants of our approach. For each table, we report EM, the individual evaluation scores from all three LLM judges, as well as their average, enabling a comprehensive assessment of evaluator agreement.

Across all baseline and ablation settings, LLM-based evaluation consistently assigns higher scores than EM, highlighting its ability to capture semantic equivalence beyond rigid string matching. More importantly, despite minor variations in absolute scores among different evaluators, the relative ranking of methods remains highly consistent across Qwen2-72b, DeepSeek-v3, and GPT-4. This observation holds for both comparisons against strong baselines and controlled ablation variants.

Such cross-evaluator stability demonstrates that the observed performance improvements of our framework are not artifacts of a specific evaluator model. Instead, they are consistently supported by multiple independently trained large language models, providing strong evidence for the robustness and reliability of the reported gains.

*Table 5.* Evaluation results on AITQA using EM and LLM-based evaluators.The LLM evaluation average is computed over Qwen2-72b, DeepSeek-v3, and GPT-4 evaluators.

| Method | EM | Qwen2-72b | DeepSeek-v3 | GPT-4 | Avg. |
|---|---|---|---|---|---|
| Chain-of-Table (Qwen2-72b) | 49.32 | 61.04 | 62.14 | 62.88 | 62.02 |
| E5 (Qwen2-72b) | 56.31 | 59.13 | 58.73 | 59.04 | 58.97 |
| St-RAPTOR (Qwen2-72b) | 60.58 | 71.29 | 70.70 | 71.09 | 71.03 |
| Ours (Qwen2-72b) | 69.34 | 89.25 | 89.25 | 88.86 | 89.12 |
| TableLLaMA-7B | 68.35 | 85.93 | 85.55 | 85.35 | 85.61 |
| Ours (TableLLaMA-7B) | 73.78 | 87.89 | 88.06 | 87.89 | 87.95 |

*Table 6.* Evaluation results on Hitab using EM and LLM-based evaluators.The LLM evaluation average is computed over Qwen2-72b, DeepSeek-v3, and GPT-4 evaluators.

| Method | EM | Qwen2-72b | DeepSeek-v3 | GPT-4 | Avg. |
|---|---|---|---|---|---|
| Chain-of-Table (Qwen2-72b) | 44.26 | 62.69 | 63.25 | 62.83 | 62.92 |
| E5 (Qwen2-72b) | 43.56 | 47.16 | 48.28 | 48.35 | 47.93 |
| St-RAPTOR (Qwen2-72b) | 53.85 | 60.35 | 60.72 | 61.07 | 60.71 |
| Ours (Qwen2-72b) | 60.16 | 66.83 | 66.79 | 67.82 | 67.15 |
| TableLLaMA-7B | 64.71 | 67.30 | 66.73 | 66.95 | 66.99 |
| Ours (TableLLaMA-7B) | 63.62 | 65.81 | 66.18 | 66.73 | 66.24 |

*Table 7.* Evaluation results on Hitab subset using EM and LLM-based evaluators. The LLM evaluation average is computed over Qwen2-72b, DeepSeek-v3, and GPT-4 evaluators.

| Method | EM | Qwen2-72b | DeepSeek-v3 | GPT-4 | Avg. |
|---|---|---|---|---|---|
| Chain-of-Table(Qwen2-72b) | 46.09 | 64.50 | 63.53 | 64.16 | 64.06 |
| E5 (Qwen2-72b) | 44.49 | 48.70 | 48.91 | 48.74 | 48.78 |
| St-RAPTOR (Qwen2-72b) | 55.73 | 61.74 | 61.95 | 62.03 | 61.91 |
| Ours (Qwen2-72b) | 64.74 | 70.25 | 70.98 | 70.75 | 70.66 |
| TableLLaMA-7B | 66.75 | 71.34 | 71.12 | 72.21 | 71.56 |
| Ours (TableLLaMA-7B) | 68.37 | 74.56 | 73.89 | 74.25 | 74.23 |

*Table 8.* Ablation study results with detailed LLM-based evaluation. The LLM evaluation average is computed over Qwen2-72b, DeepSeek-v3, and GPT evaluators.

| Variant | AITQA | | | | | HiTab | | | | |
|---|---|---|---|---|---|---|---|---|---|---|
| | EM | Qwen2-72b | DeepSeek-v3 | GPT | Avg. | EM | Qwen2-72b | DeepSeek-v3 | GPT | Avg. |
| w/o $\mathbb{C}_{\text{semantic}}$ | 63.30 | 84.18 | 84.57 | 83.97 | 84.24 | 53.72 | 59.42 | 60.18 | 60.10 | 59.90 |
| w/o $\mathcal{T}_P$ ($\mathcal{T}_{col}$) | 49.13 | 68.36 | 68.75 | 68.72 | 68.61 | 56.88 | 62.10 | 62.84 | 62.74 | 62.56 |
| w/o $\mathcal{T}_P$ ($\mathcal{T}_{row}$) | 60.78 | 85.74 | 86.33 | 85.93 | 86.00 | 56.44 | 62.92 | 63.51 | 63.47 | 63.30 |
| w/o Association (Markdown) | 53.20 | 59.96 | 60.35 | 59.93 | 60.08 | 53.35 | 57.82 | 58.44 | 58.34 | 58.20 |
| w/o Association (HTML) | 50.10 | 60.94 | 61.52 | 61.23 | 61.23 | 55.18 | 58.97 | 59.54 | 59.48 | 59.33 |
| w/o LLM-based Heuristics | 68.74 | 85.94 | 86.52 | 86.17 | 86.21 | 59.79 | 65.01 | 65.58 | 65.40 | 65.33 |
| **Full OHD** | **69.34** | **89.25** | **89.25** | **88.86** | **89.12** | **60.16** | **66.83** | **66.79** | **67.82** | **67.15** |

To ensure transparency and reproducibility of the LLM-based evaluation, we explicitly specify the prompt used by all evaluator models. The same prompt is shared across Qwen2-72b, DeepSeek-v3 and GPT-4,and is applied uniformly in both baseline comparisons (Tables 5 -7) and ablation studies (Table 8). For clarity, we provide the complete evaluation prompt below.

---

**LLM-based Evaluation Prompt**

**System Role:** You are a professional Table QA evaluation expert. Your task is to determine whether the model's prediction is correct by comparing the "Gold Label" with the "Prediction".

**Evaluation Principles:**

- **Semantic Consistency:** Judge as correct (1) if the prediction conveys the same meaning as the gold label, regardless of phrasing.
- **Numerical Tolerance:** Ignore formatting (e.g., commas, %). For differing decimal places, round the longer value to match the shorter one.
- **Unit Handling:** Units in the prediction do not affect judgment if the question already specifies them.
- **Output:** Output 1 for correct, 0 for incorrect. Only output the digit.

**Input Format:**
```
Question: [Question Text]
Gold Label: [Correct Answer]
Prediction: [Model Output]
```

---

## C. Prompt for the semantic constraint $\mathbb{C}_{\text{semantic}}$

The following prompt implements the semantic predicate $\mathbb{C}_{\text{semantic}}$ to determine hierarchical dependencies between two header cells in a complex table.

---

**Hierarchical Dependency Arbitration Prompt**

**Task:** Analyze the logical relationship between two header cells (Primary Header and Target Header) in a complex table to determine their hierarchical dependency.

**Input:**

- `Primary Header`(<primary>)
- `Target Header`(<target>)

**Instructions:** Evaluate the relationship between <primary>and <target>based on the following three categories:

1. **Sub-category (Logical Containment):** <target>is a finer-grained classification or a subset of <primary>.
   *Example:* "January" is a sub-category of "Q1".

2. **Attribute (Property Attachment):** <target>is a specific metric, feature, or attribute belonging to <primary>.
   *Example:* "Revenue" is an attribute of "Product A".

3. **Orthogonal (Parallel/Independent):** <primary>and <target>are independent entities at the same level or belong to different dimensions.
   *Example:* "Sales" and "Profit" are orthogonal if they are siblings.

**Constraint:** If the relationship is **Sub-category** or **Attribute**, consider it a "`Dependency Found`". Otherwise (**Orthogonal**), mark it as "`Boundary Reached`".

**Response Format:**
```
Relationship: [Sub-category / Attribute / Orthogonal]
```

---

## D. Prompt for Semantic Arbitration

The following prompt is used in the semantic arbitration and refinement to select the better textual representation between two candidates based on the given question. It also optimizes the chosen expression.

**Response Selection and Polishing Prompt**

*Given two texts and the question, please analyze which one is better suited to solve the question in terms of logical presentation of information (such as whether it is convenient for data comparison and record viewing), text readability (whether it is smooth and easy to understand), and information completeness (whether it comprehensively reflects the key information of the table). Then, optimize the expression in the direction of being more concise and logically clear.*

**Input:**
`Question, sentence A, sentence B`

**OUTPUT:**
`[sentence A / sentence B]`

These prompts are deployed in a zero-shot configuration leveraging state-of-the-art large language models (e.g., Qwen2-72B, deepseek-v3). The model serves as a semantic adjudicator, evaluating the candidate responses based on pre-defined logical and linguistic criteria to select the optimal output. Following this selection, the process transitions to an automated template-filling phase, where the chosen response is programmatically injected into the final Table QA framework. This systematic integration ensures that the generated answers maintain high factual fidelity while adhering to the structured formatting requirements of the downstream task.

