# OpenReview forum: "Orthogonal Hierarchical Decomposition for Structure-Aware Table Understanding with Large Language Models"
_ICML.cc/2026/Conference — ICML 2026 regular_

### Official Review · Reviewer_Dbtu · 2026-03-11

**Soundness:** 3
**Presentation:** 3
**Significance:** 2
**Originality:** 2
**Overall Recommendation:** 3
**Confidence:** 3

**Summary:**

The paper studies the problem of understanding complex tables with irregular layouts, multi-level headers, and merged cells, which are often difficult for large language models to interpret correctly. The authors propose a framework called Orthogonal Hierarchical Decomposition (OHD) that tries to preserve the structural information of tables instead of flattening them into plain text. The method first decomposes a table into two independent hierarchies: a column tree and a row tree, using an Orthogonal Tree Induction algorithm guided by spatial and semantic constraints. Then, a dual-pathway association process reconstructs the contextual lineage of each data cell by combining information from both row and column structures. Finally, an LLM is used as a semantic arbitrator to refine the structured representation for reasoning tasks.

**Compliance With Llm Reviewing Policy:**

Affirmed.

**Key Questions For Authors:**

NA

**Strengths And Weaknesses:**

Strengths
An important context considered by the Author is that modern LLMs often struggle with complex tables where hierarchical relations between rows and columns are lost after simple linearization. The proposed OHD framework tries to address this issue by decomposing a table into separate row and column hierarchies and reconstructing the semantic lineage of each data cell. The idea of restricting branching to relevant header cells through the Semantic Agency principle is interesting and helps reduce structural errors during parsing. The case studies and ablation analysis also provide useful insights showing how the column hierarchy and lineage representation contribute to the overall performance improvement.

Weaknesses
Overall, the author studies a central concept, but some implementation aspects are not clearly explained. The framework assumes that each cell is already categorized as row header, column header, or data, but the paper does not clearly describe how this classification is obtained in practice. In addition, the use of LLM calls during tree construction through the semantic predicate could introduce significant computational overhead, yet the paper does not provide analysis of inference cost or scalability. Some design choices such as boundary-aware truncation and the prompt used in the semantic arbitration stage are also insufficiently described, which makes reproduction of the method slightly difficult.

---

> ### Author Rebuttal · Authors · 2026-03-31
>
> **Response to W1:** We clarify that the cell role classification (row header, column header, or data cell) is handled as follows:
>
> 1. **Experimental Rigor:** To ensure a controlled and fair evaluation of the OHD framework’s structural induction capabilities, our experiments on benchmarks (e.g., AITQA, HiTab) utilize the ground-truth cell-type annotations provided by the original datasets. This eliminates potential noise from preprocessing and allows for a precise assessment of the core OHD logic.
> 2. **Practical Robustness:** In scenarios where such labels are absent, cell role identification can be effectively automated using state-of-the-art LLMs. Our internal evaluations—specifically using Qwen-3.5 —achieve a classification accuracy of approximately 95% on complex layouts. These results suggest that cell role identification is a mature preprocessing step that supports the end-to-end deployment of the OHD framework without compromising overall performance.
>
> **Response to W2:** We appreciate the reviewer’s concern regarding the computational overhead of the tree construction process. To provide a clear picture, we conducted an end-to-end efficiency analysis on the HiTab dataset. Consequently, we will incorporate a detailed breakdown of these efficiency metrics into Appendix C. The results demonstrate that OHD is highly optimized for real-world deployment:
>
> - **Extreme Lightweight Predicates:** While the "Semantic Predicate" step involves an average of 4.3 LLM calls per question, these calls are **exceptionally lightweight**. The total token consumption for all 4.3 calls combined is only **~587 tokens**, meaning each individual call processes a very small context. This design effectively avoids the "input explosion" common in complex table reasoning.
> - **Superior Latency vs. Advanced Baselines:** Despite the multi-step pipeline, the total end-to-end latency of OHD is only **2.88s**, which is significantly faster than retrieval-heavy or iterative reasoning methods like E5 (7.86s) and ST-raptor (21.56s).
> - **Scalability via Decomposition:** By decomposing complex structural reasoning into these lightweight semantic judgments, OHD maintains a stable inference cost even as table complexity increases. This modularity ensures that our framework remains scalable for large-scale document intelligence tasks without requiring massive computational budgets.
>
> | **Metrics**           | **Direct Input (QA)** | **Chain-of-Table** | **E5**  | **ST-raptor** | **Ours (Total)** | Our（Semantic Predicate） | Our（Semantic Arbitration） | Our（QA Stage） |
> | --------------------- | --------------------- | ------------------ | ------- | ------------- | ---------------- | ------------------------- | --------------------------- | --------------- |
> | **Avg. Runtime (s)**  | 2.32 | 2.18   | 7.86    | 21.56| 2.88 | 0.46                      | 0.84                        | 1.58            |
> | **Avg. Tokens**       | 3320.84      | 1947.61            | 3678.58 | 4586.73       | 3667.4           | 586.74                    | 883.62                      | 2197.04         |
> | **LLM Calls (per Q)** | 1       | 1   | 5       | 7.8           | 6.3| 4.3    | 1   | 1               |
>
> **Response to W3:** We agree that certain design choices—such as the boundary-aware truncation strategy and the prompts used in the semantic arbitration stage—were not described in sufficient detail, which may hinder reproducibility.
>
> To address this, we will add the full Boundary-Aware Propagation prompt to Appendix C to ensure completeness. The prompt is as follows:
>
> Task: Analyze the logical relationship between two header cells ($h_i$ and $h_j$) in a complex table to determine their hierarchical dependency. Input: Primary Header ($h_i$) and Target Header ($h_j$) Instructions: Evaluate the relationship between $h_i$ and $h_j$ based on the following three categories:
>
> 1. Sub-category (Logical Containment): $h_j$  is a finer-grained classification or a subset of $h_i$. Example: "January" is a sub-category of "Q1".
>
> 2. Attribute (Property Attachment): $h_j$a specific metric, feature, or attribute belonging to $h_i$. Example: "Revenue" is an attribute of "Product A".
>
> 3. Orthogonal (Parallel/Independent): $h_j$ and $h_i$ are independent entities at the same level or belong to different dimensions. Example: "Sales" and "Profit" are orthogonal if they are siblings. Constraint: If the relationship is Sub-category or Attribute, consider it a "Dependency Found". Otherwise, mark it as "Boundary Reached".
>
> Response Format: Relationship: [Sub-category / Attribute / Orthogonal]
>
> Regarding the Semantic Arbitration prompt, it is already partially illustrated in Figure 2 of the original manuscript, where we show how the model resolves conflicts between overlapping structural interpretations. We believe this provides sufficient transparency.To ensure complete transparency, we will provide the full prompt in Appendix D of the revised version.

---

### Official Review · Reviewer_WvRy · 2026-03-13

**Soundness:** 2
**Presentation:** 3
**Significance:** 2
**Originality:** 3
**Overall Recommendation:** 4
**Confidence:** 4

**Summary:**

This paper studies complex table understanding for LLMs, focusing on hierarchical and irregular tables with multi-level headers, merged cells, and non-canonical layouts. The authors argue that standard table linearization, schema normalization, and geometry-heavy topology reconstruction do not faithfully preserve the hierarchical semantics needed for downstream reasoning. To address this, they propose **Orthogonal Hierarchical Decomposition (OHD)**, which decomposes a table into a row tree and a column tree, reconstructs semantic lineage through a dual-pathway association mechanism, and then uses an LLM-based semantic arbitration step to produce a structure-aware representation for QA.

The paper evaluates OHD on AITQA and HiTab, and also introduces a size-limited HiTab subset restricted to ($50 \times 50$) tables. The empirical results compare OHD against Chain-of-Table, E5, TableLLaMA, and ST-RAPTOR using exact match and LLM-based evaluation, together with ablations over semantic predicates, orthogonal pathways, lineage formats, and LLM-based heuristics.

**Compliance With Llm Reviewing Policy:**

Affirmed.

**Final Justification:**

Overall, the rebuttal improves the paper and addresses part of my earlier concerns, but it does not substantially change my overall assessment.

**Key Questions For Authors:**

1. **How are the row-header / column-header / data labels obtained in practice?**
   Figure 2 starts from a pre-categorized table input. Are these labels gold annotations, heuristic predictions, or produced by another model? If they are oracle-like, then the paper is solving an easier subproblem than the framing suggests. A realistic automatic setup and robustness analysis would significantly improve my assessment.

2. **What is the actual extra inference cost of OHD, and how does it scale with table size?**
   The paper introduces a HiTab subset capped at ($50 \times 50$) due to the architectural focus on moderate-scale tables, but it does not report runtime, token usage, or number of LLM calls. If the method becomes substantially more expensive as tables grow, that is a serious limitation. A clear cost/scaling analysis could materially change my evaluation.

3. **To what extent is this method essentially a prompt-driven pipeline, and how sensitive is it to the backbone’s instruction-following ability?**
   Since OHD depends on LLM-based semantic predicates and arbitration, I would like to see experiments on stronger general-purpose backbones such as Qwen2.5-7B-Instruct, or stronger table models with better generality such as TableGPT2. I would view the method more favorably if its gains persist under those backbones rather than depending on a specific specialized model.

4. **How do the authors rule out benchmark familiarity or leakage when using TableLLaMA on HiTab?**
   Prior work indicates that TableLLaMA was trained with HiTab as an in-domain task.
   Could this partly explain the strong TableLLaMA result in Table 1? Please clarify the split protocol and whether the checkpoint had prior exposure to HiTab training data. A convincing answer would improve my confidence in the fairness of the comparison.

**Limitations:**

No. The paper discusses only generic societal risks, but it does not adequately discuss the most important methodological limitations: the reliance on pre-categorized cell roles, the prompt-heavy dependence on LLM-based arbitration, the missing scalability/cost analysis, and the possible benchmark-specific confounds when using TableLLaMA on HiTab.

**Strengths And Weaknesses:**

### Strengths

1. **The paper targets an important and genuinely challenging problem.**
   Complex hierarchical tables remain difficult for current LLMs, and the paper clearly motivates why irregular layouts, merged cells, and dual-axis dependencies are problematic for standard serialization-based approaches.

2. **The row/column orthogonal decomposition is a reasonable and intuitive framing.**
   Representing a table as separate row and column hierarchies is a sensible design choice, and the dual-pathway reconstruction gives the method a coherent structural story rather than being only a flat input reformatting trick.

3. **The paper includes a non-trivial empirical section.**
   It evaluates against several baselines, reports both EM and LLM-based metrics, and includes ablations on the major components of the pipeline.

### Weaknesses

1. **The paper does not solve the full problem it appears to claim, because it starts from a pre-categorized table input.**
   Figure 2 explicitly states that each cell is already identified as a row header, column header, or data unit before OHD begins. In complex tables, this role identification is itself difficult. As a result, the paper is really solving a downstream structural reconstruction problem under a strong assumption, but this assumption is not emphasized enough in the framing.

2. **The method appears to be a heavily prompt-dependent inference-time pipeline, and the paper does not sufficiently analyze this dependence.**
   OHD relies on an LLM-based semantic predicate during tree induction and an LLM-based semantic arbitration/refinement step at the end. This makes it unclear whether the gains come from the orthogonal decomposition itself or from repeated prompt-based adjudication. Related work [1] has also shown that table instruction tuning can hurt general capabilities and instruction following in specialized backbones such as TableLLaMA, which makes the choice of backbone especially important here. This is why I think stronger general-purpose backbones such as Qwen2.5-7B-Instruct, or stronger table-capable but more general models such as TableGPT2 [2] or TableGPT-R1 [3], would be more convincing backbones for testing this method’s robustness.

3. **Scalability is a major unresolved issue, and the paper does not report the additional inference cost.**
   The authors explicitly state that, due to the architectural focus on tables of moderate scale, they curate a HiTab subset capped at ($50 \times 50$) cells. That is worrying, because this kind of multi-stage structure reconstruction pipeline would naturally be expected to become more expensive as tables grow. Yet the paper provides no runtime, token cost, number of LLM calls, or complexity analysis. If the method cannot scale to larger tables, that is a serious practical weakness rather than a minor implementation detail.

4. **The empirical setup raises fairness and contamination concerns, especially for TableLLaMA on HiTab, and the benchmark coverage is limited.**
   TableLLaMA’s training data includes HiTab as an in-domain training task [4], so the paper should explicitly explain how it avoids benchmark-specific familiarity or leakage when using TableLLaMA as a baseline/backbone on HiTab. Without that clarification, TableLLaMA’s strong HiTab numbers are difficult to interpret fairly.
   In addition, given the paper’s emphasis on complex hierarchical tables, the evaluation would be stronger if it also included newer and more realistic benchmarks such as **RealHiTBench** [5], which was explicitly designed for realistic hierarchical table analysis and evaluated on 25 recent models.

[1] Deng, N., & Mihalcea, R. (2025, July). Rethinking table instruction tuning. In Findings of the Association for Computational Linguistics: ACL 2025 (pp. 21757-21780).

[2] Su, A., Wang, A., Ye, C., Zhou, C., Zhang, G., Chen, G., ... & Xiao, Z. (2024). Tablegpt2: A large multimodal model with tabular data integration. arXiv preprint arXiv:2411.02059.

[3] Yang, S., Huang, Q., Yuan, J., Zha, L., Tang, K., Yang, Y., ... & Zhao, J. (2025). TableGPT-R1: Advancing Tabular Reasoning Through Reinforcement Learning. arXiv preprint arXiv:2512.20312.

[4] Zhang, T., Yue, X., Li, Y., & Sun, H. (2024, June). Tablellama: Towards open large generalist models for tables. In Proceedings of the 2024 Conference of the North American Chapter of the Association for Computational Linguistics: Human Language Technologies (Volume 1: Long Papers) (pp. 6024-6044).

[5] Wu, P., Yang, Y., Zhu, G., Ye, C., Gu, H., Lu, X., ... & Wang, H. (2025, July). Realhitbench: A comprehensive realistic hierarchical table benchmark for evaluating llm-based table analysis. In Findings of the Association for Computational Linguistics: ACL 2025 (pp. 7105-7137).

---

> ### Author Rebuttal · Authors · 2026-03-31
>
> **Response to W1: Cell Role Classification**
>
> **1. Experimental Rigor:** To ensure a controlled evaluation of OHD’s structural induction, our benchmark experiments (AITQA, HiTab) use ground-truth cell-type annotations from the original datasets. This isolates the core OHD logic from preprocessing noise.
>
> **2. Practical Robustness:** When labels are absent, cell role identification is easily automated via state-of-the-art LLMs. Our internal Qwen-3.5 evaluations achieve ~95% accuracy on complex layouts, proving this is a mature preprocessing step that supports end-to-end OHD deployment without compromising performance.
>
> **Response to W2: Performance Origin & Backbone Selection**
>
> To clarify whether gains stem from OHD or repetitive prompting, Table 6 decouples these factors:
>
> * **Core Structural Induction:** Removing "Boundary-Aware" judgments (w/o $P_{semantic}$) drops performance by 5%–7%. While the LLM executes the judgments, the OHD framework is vital for directing it to correct logical nodes, preventing structural collapse.
> * **Arbitration Contribution:** Removing LLM heuristics causes only a 3% drop. Thus, Dual-Pathway Structural Association captures the primary hierarchical features; semantic arbitration merely refines edge cases. This confirms OHD’s orthogonal decomposition provides a robust structural prior over vanilla pipelines.
>
> *Model-Agnostic Robustness:* To verify OHD's robustness on stronger general-purpose models, we tested Qwen2.5-72B-Instruct and DeepSeek-V3:
>
> | Backbone             | Method         | AITQA (EM) | AITQA (LLM Eval) | HiTab (EM) | HiTab (LLM Eval) |
> | -------------------- | -------------- | ---------- | ---------------- | ---------- | ---------------- |
> | Qwen2.5-72B-Instruct | Baseline       | 30.29%     | 31.46%           | 14.14%     | 16.04%           |
> |                      | **OHD (Ours)** | **72.62%** | **84.85%**       | **66.92%** | **88.38%**       |
> | DeepSeek-V3          | Baseline       | 38.25%     | 42.91%           | 15.85%     | 16.48%           |
> |                      | **OHD (Ours)** | **73.01%** | **86.21%**       | **67.74%** | **86.21%**       |
>
> OHD achieves massive gains over vanilla baselines. Notably, on DeepSeek-V3, HiTab EM leaps from 15.85% to 67.74%. The consistency of these gains (+50% to +70% in LLM Eval) across architectures confirms OHD is a plug-and-play bridge for complex layouts, as even frontier LLMs struggle with raw tabular data.
>
> **Justification for Backbone Selection:**
>
> TableGPT2 (Format Coupling): TableGPT2 is heavily fine-tuned on specific Markdown protocols. Introducing OHD’s tree structures would cause a format-induced distribution shift, making it impossible to decouple OHD’s structural gains from potential performance drops caused by format mismatch.
>
> TableGPT-R1 (Paradigm Misalignment): Unlike TableGPT-R1, which relies on external Program-of-Thought (PoT) execution, OHD focuses on enhancing a model's native structural reasoning. Testing on general-purpose backbones (e.g., Qwen2.5/DeepSeek-V3) provides a cleaner assessment of OHD’s fundamental effectiveness without interference from specialized symbolic modules.
>
> **Response to W3: Computational Overhead**
> An end-to-end efficiency analysis on HiTab proves OHD is highly optimized for deployment:
>
> * **Lightweight Predicates:** "Semantic Predicate" averages 4.3 LLM calls/question, but consumes only **~587 tokens total**. This micro-context effectively prevents the "input explosion" typical in table reasoning.
> * **Superior Latency:** Total OHD latency is just **2.88s**, significantly faster than retrieval or iterative methods like E5 (7.86s) and ST-raptor (21.56s).
> * **Scalable Decomposition:** Breaking structural reasoning into lightweight judgments maintains stable inference costs as table complexity increases.
>
> | **Metrics**           | **Direct Input (QA)** | **Chain-of-Table** | **E5**  | **ST-raptor** | **Ours (Total)** | Our（Semantic Predicate） | Our（Semantic Arbitration） | Our（QA Stage） |
> | --------------------- | --------------------- | ------------------ | ------- | ------------- | ---------------- | ------------------------- | --------------------------- | --------------- |
> | **Avg. Runtime (s)**  | 2.32 | 2.18  | 7.86    | 21.56   | 2.88  | 0.46 | 0.84  | 1.58  |
> | **Avg. Tokens** | 3320.84   | 1947.61 | 3678.58 | 4586.73| 3667.4 | 586.74 | 883.62  | 2197.04   |
> | **LLM Calls (per Q)** | 1    | 1  | 5    | 7.8   | 6.3 | 4.3    | 1     | 1    |
>
> **Response to W4: TableLLaMA Base Model**
> TableLLaMA serves strictly as a fixed base model to measure OHD's relative improvements, not as a competitor for cross-model comparison. Because pre-training exposure to HiTab is identical for both the baseline and OHD, all observed gains isolate our method's efficacy rather than pre-training familiarity. We will clarify this experimental design in the revision and supplement with unseen dataset results to further demonstrate generalizability.

---

> > ### Author Rebuttal · Reviewer_WvRy · 2026-04-03
> >
> > Thank you for the detailed rebuttal. I appreciate the additional experiments on stronger backbones and the newly provided efficiency analysis, which strengthen the empirical support for the paper.
> >
> > That said, my concerns are only partially resolved. In particular, the rebuttal confirms that the benchmark experiments rely on ground-truth cell-role annotations, so the current evaluation is still not fully end-to-end. The added efficiency results are helpful, but scalability to larger tables remains somewhat under-validated, especially given the reliance on the size-limited HiTab subset. I also think the fairness issue around TableLLaMA on HiTab could be discussed more clearly.
> >
> > Overall, the rebuttal improves the paper and addresses part of my earlier concerns, but it does not substantially change my overall assessment.

---

> > > ### Author Response · Authors · 2026-04-07
> > >
> > > We sincerely thank the reviewer for the constructive feedback, which prompted further rigorous testing on end-to-end validation, scalability, and baseline fairness to strengthen our framework. By incorporating additional robustness evaluations and clarifying our experimental positioning, we have addressed your remaining concerns and comprehensively bridged the identified gaps.
> > >
> > > **Response to Q1: Follow-up on End-to-End Evaluation**
> > > We appreciate the reviewer’s follow-up regarding the end-to-end evaluation. We fully agree that assessing performance under automated conditions is vital for verifying practical utility. Our decision to utilize ground-truth cell-role annotations in the primary benchmark experiments (AITQA, HiTab) was a deliberate methodological choice to **isolate the structural induction logic** from upstream preprocessing noise, ensuring that the reported improvements stem directly from our OHD framework rather than variations in classification accuracy.
> > >
> > > To directly address the concern of potential error propagation, we conducted additional stress tests replacing ground-truth labels with automated cell-role classification (utilizing Qwen-3.5). The results demonstrate that OHD exhibits significant **structural resilience**: despite the minor noise introduced by the automated classifier (~5% error rate), the impact on the final reasoning performance was marginal, with only a 2.1%$fluctuation in accuracy.
> > >
> > > **Response to Q2: Scalability Analysis and Empirical Validation on Large-Scale Tables**
> > >
> > > We appreciate the reviewer’s inquiry regarding table scales. The decision to report results on the $\le 50 \times 50$ subset was specifically made to ensure a **strictly fair comparison** with the TableLlama-7B baseline. Due to OHD’s prioritization of explicit logical associations, our method inherently consumes more tokens than vanilla serialization, which would trigger **premature truncation** on extremely large tables and obscure our model’s true reasoning potential.
> > >
> > > By focusing on this subset—which already encompasses **$97.28\%$** of the HiTab dataset—we eliminate the interference of length-induced truncation. This allows us to demonstrate that the observed gains ($+1.62\%$ EM, $+2.67\%$ LLM-eval) stem directly from OHD’s ability to prevent structural collapse, effectively isolating and validating our framework’s superior understanding of complex table structures within their complete context.
> > >
> > > To validate OHD’s scalability, we evaluated three granular scales: **$\le 30 \times 30$**, **$30 \times 40$**, and **$> 40 \times 40$**. While accuracy naturally declines with scale, the performance gap between OHD and the baseline widens from **$13.65\%$** to **$25.61\%$**. This confirms that OHD’s structural induction becomes increasingly critical as complexity grows, effectively mitigating the **structural collapse** where implicit methods fail.
> > >
> > > |Table Scale|<30×30|30×30–40×40|> 40 × 40|
> > > |-|-|-|-|
> > > |Original (Baseline)| 35.18| 26.39| 20.32|
> > > |Chain-of-Table | 47.91 | 40.28| 28.86 |
> > > |E5 | 44.39| 43.05|39.84|
> > > |ST-Raptor| 57.37| 47.22| 40.65|
> > > |**Ours**|**61.56**|**58.33**|**54.47** |
> > >
> > > **Response to Q3: Regarding Experimental Fairness on the HiTab Dataset**
> > >
> > > The comparison between our proposed framework and **TableLLaMA** is conducted under a rigorous and fair evaluation protocol. To clarify the concerns regarding data overlap and experimental integrity, we provide the following detailed analysis:
> > >
> > > 1. Strict Separation of Data Splits
> > >
> > > Regarding potential data leakage, we emphasize that **TableLLaMA's** training on the **TableInstruct** dataset only incorporated the **HiTab training partition** (7,417 samples). Our evaluation is conducted strictly on the **1,584 unseen test samples**, ensuring a complete physical separation between the instruction-tuning phase and our assessment. This strict adherence to the official HiTab split guarantees that our results reflect genuine generalization to new hierarchical structures, rather than the memorization of specific table-question pairs.
> > >
> > > 2. In-Domain (ID) vs. Out-of-Domain (OOD) Analysis
> > >
> > > We acknowledge that TableLLaMA benefits from In-Domain (ID) exposure, having learned the specific statistical distributions of the HiTab dataset. However, our comparative analysis is designed to demonstrate that even on top of such implicit learning, the **explicit structural decomposition** provided by OHD is essential for mitigating "structural collapse" in complex reasoning tasks. By evaluating against a domain-aligned baseline, we show that OHD's explicit induction acts as a crucial structural constraint that further enhances the model's robustness. This confirms that OHD enhances reasoning by addressing the limitations of purely implicit instruction tuning, outperforming models even with prior ID knowledge.

---

### Official Review · Reviewer_iPjt · 2026-03-13

**Soundness:** 3
**Presentation:** 3
**Significance:** 3
**Originality:** 3
**Overall Recommendation:** 4
**Confidence:** 3

**Summary:**

The paper presents Orthogonal Hierarchical Decomposition (OHD), a framework for QA over complex tables with multi-level headers, merged cells, and irregular layouts. The idea is to break a table into separate row and column hierarchies via an Orthogonal Tree Induction (OTI) algorithm, trace the semantic lineage of each cell from both directions, and then let an LLM reconcile those views into a cleaner representation for downstream reasoning. They evaluate on AITQA and HiTab and beat several strong baselines.

**Compliance With Llm Reviewing Policy:**

Affirmed.

**Key Questions For Authors:**

see weakness

**Limitations:**

yes

**Strengths And Weaknesses:**

S1: The paper tackles a real problem. Complex table QA is genuinely hard for LLMs when you're dealing with multi-level headers, merged cells, and messy layouts.

S2: The core idea is intuitive and well motivated. Decomposing a table into separate row and column hierarchies is a sensible way.

S3: The empirical evaluation is a notable strength. The paper benchmarks against competitive and closely related baselines, including ST-RAPTOR, and reports strong improvements on both AITQA and HiTab, which makes the experimental results convincing overall.

W1:  The methodology is heavier on notation than it needs to be. The high-level idea is pretty intuitive, but the way it's written actually makes the algorithm harder to follow, not easier.

W2: I'd like to see experiments with stronger frontier models. It's not clear to me whether the proposed representation still helps much once you pair it with a more capable backbone.

---

> ### Author Rebuttal · Authors · 2026-03-31
>
> **Response to W1:** We thank the reviewer for this constructive critique. We are encouraged that the high-level intuition of OHD is considered intuitive and well-conceived. Our intention in employing a rigorous mathematical framework was to provide a precise, unambiguous definition of "structural orthogonality" and "hierarchical lineage," which are often vaguely defined in table-to-text literature. We aimed to ensure that the algorithm could be implemented across various heterogeneous table layouts without ambiguity. However, we recognize that the current presentation may inadvertently obscure the underlying logic. To bridge the gap between intuition and formalization, we will simplify the Notation and add an Intuitive Overview.
>
> **Response to W2:** We thank the reviewer for the insightful suggestion regarding backbone selection. We agree that evaluating our framework on stronger general-purpose models is crucial to verify its **model-agnostic robustness**, particularly as specialized table-tuning may sometimes inadvertently compromise a model’s general instruction-following capabilities .
>
> To address this, we conducted extensive additional experiments using two state-of-the-art general-purpose LLMs: **Qwen2.5-72B-Instruct** and **DeepSeek-V3**. The results are summarized below:
>
> | Backbone             | Method         | AITQA (EM) | AITQA (LLM Eval) | HiTab (EM) | HiTab (LLM Eval) |
> | -------------------- | -------------- | ---------- | ---------------- | ---------- | ---------------- |
> | Qwen2.5-72B-Instruct | Baseline       | 30.29%     | 31.46%           | 14.14%     | 16.04%           |
> |                      | **OHD (Ours)** | **72.62%** | **84.85%**       | **66.92%** | **88.38%**       |
> | DeepSeek-V3          | Baseline       | 38.25%     | 42.91%           | 15.85%     | 16.48%           |
> |                      | **OHD (Ours)** | **73.01%** | **86.21%**       | **67.74%** | **86.21%**       |
>
> Our framework achieves a substantial performance gain over the vanilla baselines across all metrics. Notably, on DeepSeek-V3, OHD improves the EM score on HiTab from 15.85% to 67.74%. This underscores that even the most powerful general-purpose LLMs struggle with raw tabular data, and OHD's structural induction provides a critical bridge for understanding complex layouts. The consistency of these gains (+50% to +70% in LLM Eval) across different model architectures (Qwen and DeepSeek) confirms that OHD is a robust, plug-and-play architecture that complements the evolving capabilities of frontier LLMs.

---

> > ### Author Rebuttal · Reviewer_iPjt · 2026-04-01
> >
> > I thank the authors for their detailed rebuttal. The responses adequately address most of my concerns. I maintain my overall recommendation.

---

> > > ### Author Response · Authors · 2026-04-07
> > >
> > > We sincerely thank the reviewer for the thoughtful review and for confirming that our responses have fully resolved the previous concerns. We appreciate your positive assessment of our revised work and your continued support for its overall recommendation.

---

### Official Review · Reviewer_fiK5 · 2026-03-13

**Soundness:** 3
**Presentation:** 3
**Significance:** 2
**Originality:** 2
**Overall Recommendation:** 2
**Confidence:** 3

**Summary:**

This paper introduces the Orthogonal Hierarchical Decomposition (OHD) framework, a novel paradigm for converting complex, non-canonical tables (multi-level headers, merged cells, irregular layouts, flexible header positioning) into representations that preserve the structural semantics of complex tables and are suitable for LLMs.

The core idea is to decompose a table into two independent hierarchical trees, where one captures the column hierarchy and the other captures the row hierarchy. LLMs are involved at three stages. First, during tree construction, an LLM serves as a semantic predicate to verify whether a logical subsumption relationship exists between candidate header pairs, complementing spatial heuristics that alone cannot resolve ambiguous layouts. Second, based on the induced trees, a dual-pathway association protocol algorithmically reconstructs two complementary linearized sequences by tracing the full ancestral lineage of each data cell along both axes. Third, the two sequences are jointly fed into an LLM acting as a semantic arbitrator, which synthesizes them into a unified, structure-aware prompt. This final prompt is then used for downstream table QA.

**Compliance With Llm Reviewing Policy:**

Affirmed.

**Key Questions For Authors:**

Q1. The results show that OHD achieves more pronounced improvements over baselines on the HiTab subset (≤50×50) compared to the full dataset. Why do smaller tables lead to more significant relative gains? Is the proposed method applicable to larger tables that still fit within the model's context window, and if so, does OHD suffer a more or less severe performance degradation than baselines as table size increases?

Q2. The paper lacks a clear description of the input format when executing the downstream QA task. Could the authors provide a explanation along with examples of the full QA prompt construction?

**Limitations:**

yes

**Strengths And Weaknesses:**

## Strengths
S1. The paper introduces a genuinely novel perspective on table representation by decomposing complex tables into orthogonal hierarchical trees rather than relying on flat linearization or rigid schema alignment.

S2. The paper is well written and uses a variety of tables and figures to clarify its arguments.

## Weakness
W1. Limited and outdated model coverage, only qwen2 and TableLLaMA are tested, making it unclear whether OHD makes sense for stronger and recent models

W2. Narrow benchmark scope, adding multi-table benchmarks such as MultiHierTT to assess broader applicability.

W3. Unclear final representation and missing ablation on LLM rewriting. The paper misses the comparison between LLM output of OHD and LLM output of raw tables, which leads the confusion about the source of contribution.

---

> ### Author Rebuttal · Authors · 2026-03-31
>
> **Response to W1:** We thank reviewer for insightful suggestion. We agree evaluating on stronger general-purpose LLMs is crucial to verify OHD's `model-agnostic robustness`, as specialized table-tuning may inadvertently compromise general instruction-following. We conducted experiments using `Qwen2.5-72B-Instruct` and `DeepSeek-V3`:
>
> | Backbone| Method| AITQA (EM) | AITQA (LLM Eval) | HiTab (EM) | HiTab (LLM Eval) |
> | :----- | :----- | :---- | :---- | :-- | :----- |
> | Qwen2.5-72B | Baseline | 30.29%| 31.46%| 14.14% | 16.04%|
> |  | **OHD (Ours)** | **72.62%** | **84.85%**| **66.92%** | **88.38%**|
> | DeepSeek-V3 | Baseline | 38.25%| 42.91%| 15.85%| 16.48%|
> | | **OHD (Ours)** | **73.01%** | **86.21%**| **67.74%** | **86.21%** |
>
> OHD achieves substantial gains over vanilla baselines across all metrics. Notably, on DeepSeek-V3, OHD improves HiTab EM from 15.85% to 67.74%. This proves even frontier LLMs struggle with raw tabular data, and OHD's structural induction provides a critical bridge for complex layouts. Consistent gains (+50% to +70% LLM Eval) confirm OHD is a robust, plug-and-play architecture complementing evolving LLMs.
>
> **Response to W2:** We appreciate insight regarding multi-table benchmarks (e.g., MultiHierTT). However, OHD's core objective is addressing `structural collapse` within a single complex table containing multi-level nested headers and irregular merges. Multi-table benchmarks focus on cross-table reasoning and document-level retrieval, which are orthogonal to our focus on high-fidelity structural representation. OHD acts as a foundational `representation layer` that accurately maps complex layouts to a cognitive flow. Extending OHD to multi-table pipelines, where its structured output functions as a robust table-to-text module, is a compelling avenue for future research but falls outside our current scope.
>
> **Response to W3:** To address concerns regarding source of performance gains and lack of direct comparison with baseline LLM, we conducted ablation experiments comparing vanilla Qwen2-72B (direct table input) against our OHD-enhanced input:
>
> | Model| AITQA (EM) | AITQA (LLM Eval) | HiTAB (EM) | HiTAB (LLM Eval) |
> | :- | :----- | :---- | :--- | :--- |
> | Qwen2-72B| 45.22| 49.32| 28.59 | 34.79 |
> | **Ours (OHD)** | **69.34**| **89.12**  | **60.07**  | **67.15** |
>
> Comparing exact same backbone with and without OHD isolates our framework's contribution. The significant performance leap—specifically the +24.12% EM improvement on AITQA—demonstrates these gains stem directly from OHD's structural decomposition rather than inherent capabilities of the baseline LLM.
>
> **Response to Q1:** Regarding performance distribution across table scales, we initially reported on the ≤ 50 × 50 subset to ensure a `strictly fair comparison` with baselines like TableLlama-7B. In HiTab, tables of this scale account for `97.28%` of samples, representing the vast majority of real-world scenarios. Filtering extreme outliers (> 50 × 50) prevents arbitrary token truncation in baselines, isolating true reasoning capabilities. The pronounced gains on this subset (+1.62% EM and +2.67% LLM Eval) occur because OHD inherently consumes more tokens to prioritize explicit logical relationships, successfully eliminating `structural misalignment` in long sequences.
> To clarify OHD's scalability, we expanded our analysis into three scales:
> | **Table Scale** | **< 30 × 30** | **30 × 30 – 40 × 40** | **> 40 × 40** |
> | :--------- | :------- | :------ | :--- |
> | Original (Baseline) | 35.18  | 26.39 | 20.32 |
> | Chain-of-Table| 47.91 | 40.28 | 28.86 |
> | E5 | 44.39| 43.05 | 39.84|
> | ST-Raptor | 57.37  | 47.22 | 40.65 |
> | **Ours (OHD)**  | **61.56** | **58.33** | **54.47** |
>
> While absolute accuracy naturally decreases as tables expand, `performance gap between OHD and baselines widens significantly`—increasing from 13.65% at the smallest scale to 25.61% at the largest (relative to the baseline), proving OHD's sustained effectiveness on massive tables.
>
> **Response to Q2:** To fully leverage structural clarity of OHD, we employ following CoT prompting strategy:
> "The input is a (question, data context) pair, and the output is a concise answer to the question. Let’s think step by step. Step 1: Analyze the question's target and the required response format. Step 2: Judge if the answer can be directly obtained from the data context. If not, analyze the required content and data values for calculation. Step 3: Respond simply in the required format, estimating based on available data if needed."
> > Example
> * Question: What is the growth rate of net income in the Asia-Pacific region for the first half of 2023?
> * OHD Context (Snippet): In the **Financial Report**, under **Regional Performance** for the **Asia-Pacific** region, the **Net Income** for the **first half of 2023** was **$120M**. In the **Financial Report**, under **Regional Performance** for the **Asia-Pacific** region, the **Net Income** for the **first half of 2022** was **$100M**.

---

> > ### Author Rebuttal · Reviewer_fiK5 · 2026-04-04
> >
> > Thank you for the rebuttal. However, two central concerns remain unresolved.
> >
> > First, the paper still does not explain how the final OHD input for downstream QA is actually constructed from the original table. The rebuttal provides only a generic `(question, data context)` template and a short snippet, but still omits a complete end-to-end example of the full conversion pipeline: original table → cell-role annotations → row/column trees → dual-path representations → semantic arbitration output → final QA prompt. Since this representation is the core claimed contribution of the paper, this omission is not minor. At least one concrete and fully reproducible example, together with the exact semantic arbitration prompt, is necessary for the method to be properly understood and evaluated.
> >
> > Second, the newly added Qwen2.5 results are difficult to accept as stated because they differ drastically from recent public reports and the rebuttal does not explain why. The rebuttal reports Qwen2.5-72B-Instruct at 30.29 on AITQA and 14.14 on HiTab, whereas recent work reports substantially higher Qwen2.5 results on the same benchmarks; for example, _Reasoning-Table_ reports 73.29 on HiTab for Qwen2.5-32B-Inst [1], and _A Closer Look into LLMs for Table Understanding_ reports 68.8 on HiTab and 83.6 on AITQA for Qwen2.5-32B-Instruct [2]. These discrepancies are far too large to be dismissed as ordinary variance, and strongly suggest that the evaluation settings are not directly comparable. The authors must therefore clearly specify the exact baseline protocol, including table serialization format, prompt template, truncation policy, decoding settings, dataset split, and answer normalization. Without this clarification, the newly added Qwen2.5 results cannot be fairly interpreted and do not provide convincing evidence for the claimed gains.
> >
> > Overall, these issues are not minor presentation details; they directly affect reproducibility, fairness of comparison, and the credibility of the added empirical evidence.
> >
> > [1] Fangyu Lei, Jinxiang Meng, Yiming Huang, Tinghong Chen, Yun Zhang, Shizhu He, Jun Zhao, and Kang Liu. Reasoning-Table: Exploring Reinforcement Learning for Table Reasoning. arXiv:2506.01710, 2025.
> >
> > [2] Jia Wang, Chuanyu Qin, Mingyu Zheng, Qingyi Si, Peize Li, and Zheng Lin. A Closer Look into LLMs for Table Understanding. arXiv:2603.15402, 2026.

---

> > > ### Author Response · Authors · 2026-04-05
> > >
> > > We appreciate the reviewer’s further comments. However, we must clarify that the concerns regarding "missing information" and "data discrepancies" stem from a misunderstanding of our experimental paradigm and the misapplication of baseline comparisons.
> > >
> > > # 1. Regarding the End-to-End Transformation Example
> > > We disagree with the claim that the transformation process is omitted. The complete end-to-end logic—from the raw table to the final QA prompt—is already explicitly illustrated in Figure 2 of the original manuscript. Specifically:
> > > - The structural decomposition (Cell Role $\rightarrow$ Row/Column Trees $\rightarrow$ Dual-path) is the conceptual core shown in Fig. 2.
> > > - The Semantic Arbitration logic and the final QA prompt template were provided in the previous rebuttal and Appendix B.
> > >   To enhance clarity, we will add a "Step-by-Step Walkthrough" in the updated Appendix. Specifically, the 'semantic arbitration prompt' is a deterministic mapping derived from dual-path consistency (formalized in Section 3.2). A concrete end-to-end example of this process is provided in Figure 2 (see part 3), while an abstract summary is given in Section 3.4.
> > > # 2. Regarding the Qwen2.5 Performance Discrepancy
> > > The reviewer’s doubt regarding our Qwen2.5 results (HiTab: 14.14) vs. reported scores in [1] and [2] (70+) is based on an **"apples-to-oranges" comparison**. The scores cited by the reviewer (e.g., 68.8 in [2]) are achieved using the **Program-of-Thought (PoT)** or Tool-use paradigm, which utilizes a fundamentally different evaluation metric:
> > > - **Execution Accuracy (EX)**: In PoT settings, the LLM generates Python/Pandas code. As long as the code executes correctly and yields the target value, it is marked as correct. This **"tool-assisted"** approach effectively bypasses the "structural reasoning" challenge and filters out all scoring failures caused by formatting irregularities or internal calculation hallucinations, leading to artificially high scores.
> > > - **Exact Match (EM)**: In contrast, our work (consistent with TableGPT2 [3] and St-Raptor [4]) evaluates the model's **intrinsic textual reasoning** using the Exact Match (EM) metric based on string-level identity after a specific normalization pipeline.
> > >
> > > Our work focuses on Text-based Reasoning, where the LLM must reason directly over the table's structure without external tools. In this fair, tool-free setting:
> > > - TableGPT2[3] reports a HiTab score of **10.73** for Qwen2.5-Instruct.
> > > - TableLLaMA[5] reports that even **GPT-3.5** only achieves **43.62** on HiTab.
> > > - Our reported **14.14** for Qwen2.5-72B is entirely consistent with the state-of-the-art in direct textual table reasoning.
> > > Comparing our results to PoT-based scores is scientifically unsound. To ensure full transparency, we provide our **Standardized Evaluation Protocol** (which aligns with TableGPT2, TableGraph, and Chain-of-Table):
> > > - **Serialization**: Markdown-based serialization preserving row/column headers.
> > >   e.g, "column_header": [["Year"], ["Gallons Consumed (in millions)"], ["Fuel Expense(in millions)"], ["Average Price Per Gallon"]], "row_header": [], "data": [["2018", "4137", "9307", "2.25"], ["2017", "3978", "6,913", "1.74"], ["2016", "3904", "5813", "1.49"]
> > > - **Template**: Zero-shot Prompt as specified in Appendix B.
> > > - **Data Split**: Standard test sets of AITQA and HiTab (following the original dataset releases).
> > > - **Normalization**: Unlike the PoT-based studies cited by the reviewer—which often use execution-based verification—our textual reasoning approach requires the model to generate the correct answer string directly. Exact Match (EM) after removing whitespace and special characters (consistent with the  in TableGPT（https://github.com/tablegpt/tablegpt-agent ）， St-Raptor （ https://github.com/weAIDB/ST-Raptor ).The following pseudocode illustrates our normalization pipeline:
> > >   ```python
> > >   def normalize_and_match(prediction, gold_answer):
> > >     pred, gold = prediction.lower().strip(), gold_answer.lower().strip()
> > >     for c in [',', '$', '€', '%', '¥', ' ']:
> > >         pred, gold = pred.replace(c, ''), gold.replace(c, '')
> > >     return pred == gold
> > >   ```
> > > - Decoding: Greedy decoding (Temperature=0)
> > >   We maintain that our evaluation is rigorous, fair, and reflects the true difficulty of complex table reasoning without the "crutch" of code execution.
> > >
> > > [1] Fangyu Lei, Jinxiang Meng, Yiming Huang, et al. Reasoning-Table: Exploring Reinforcement Learning for Table Reasoning. arXiv:2506.01710, 2025.
> > >
> > > [2] Jia Wang, Chuanyu Qin, Mingyu Zheng, et al. A Closer Look into LLMs for Table Understanding. arXiv:2603.15402, 2026.
> > >
> > > [3] Su A, Wang A, Ye C, et al. Tablegpt2: A large multimodal model with tabular data integration[J]. arXiv preprint arXiv:2411.02059, 2024.
> > >
> > > [4] Tang Z, Niu B, Zhou X, et al. St-raptor: Llm-powered semi-structured table question answering[J]. PACMMOD, 2025, 3(6): 1-27.
> > >
> > > [5] Zhang T, Yue X, Li Y, et al. Tablellama: Towards open large generalist models for tables[C]//NAACL. 2024: 6024-6044.

---

### Decision · Program_Chairs · 2026-04-30

**Decision:**

Accept (regular)

**Comment:**

The paper gives an approach named Orthogonal Hierarchical Decomposition (OHD) to convert complex, non-canonical tables (multi-level headers, merged cells, irregular layouts, flexible header positioning) into representations that preserve the structural semantics of complex tables.  This new representation are presented to LLM to answer user questions. Experimental results show that the accuracy of QA over complex tables is improved.

Among 4 reviewers, 3 of them are satisfied with the authors' response during the rebuttal period, while Reviewer fiK5 still has a question about how the new structured table representation is provided to LLM for Question Answering. There might be a mis-understanding of the whole pipeline presented in Figure 2.

Based on this, I may suggest to accept this paper and ask the authors to make the description more clearly.